# Phylogenomics and the rise of the angiosperms

Angiosperms are the cornerstone of most terrestrial ecosystems and human livelihoods[1,2]. A robust understanding of angiosperm evolution is required to explain their rise to ecological dominance. So far, the angiosperm tree of life has been determined primarily by means of analyses of the plastid genome[3,4]. Many studies have drawn on this foundational work, such as classification and first insights into angiosperm diversification since their Mesozoic origins[5–7]. However, the limited and biased sampling of both taxa and genomes undermines confidence in the tree and its implications. Here, we build the tree of life for almost 8,000 (about 60%) angiosperm genera using a standardized set of 353 nuclear genes[8]. This 15-fold increase in genus-level sampling relative to comparable nuclear studies[9] provides a critical test of earlier results and brings notable change to key groups, especially in rosids, while substantiating many previously predicted relationships. Scaling this tree to time using 200 fossils, we discovered that early angiosperm evolution was characterized by high gene tree conflict and explosive diversification, giving rise to more than 80% of extant angiosperm orders. Steady diversification ensued through the remaining Mesozoic Era until rates resurged in the Cenozoic Era, concurrent with decreasing global temperatures and tightly linked with gene tree conflict. Taken together, our extensive sampling combined with advanced phylogenomic methods shows the deep history and full complexity in the evolution of a megadiverse clade.

Flowering plants (angiosperms) represent about 90% of all terrestrial plant species[2] but, despite their remarkable diversity and ecological importance underpinning almost all main terrestrial ecosystems, their evolutionary history remains incompletely known. Since their Mesozoic origins[5,10,11], angiosperms have had a pervasive influence on the biosphere of Earth, shaping climatic changes at global and local scales[12], supporting the structure and assembly of biomes[13] and influencing the diversification of other organisms, such as insects, fungi and birds[14]. The evolution of terrestrial biodiversity is thus inextricably linked with the macroevolution of angiosperms, which can only be shown using a robust and comprehensive tree of life. Reconstructing such a tree, however, is challenging because of the sheer diversity of angiosperms and the complex phylogenetic signal in their genomes.

High-throughput DNA sequencing methods now enable us to reconstruct phylogenetic trees that broadly represent the evolutionary signal across entire genomes. Target sequence capture[15] has revolutionized plant phylogenetics by unlocking herbarium specimens as a source of sequenceable DNA[16], thus removing the chief sampling bottleneck that has obstructed the completion of the tree of life. Although previous work on plants has relied primarily on the widely sequenced plastid genome[3,4,7], these technologies now allow us to tap into the evolutionary signal of the much larger and more complex nuclear genome. Universal nuclear probe sets, such as Angiosperms353 (ref. 8), have made target sequence capture consistently applicable across broad taxonomic scales, opening doors to collaboration and data integration[17]. As a result, opportunities now present themselves to address fundamental questions in plant evolutionary biology, such as the origin of angiosperms, the tempo and mode of their diversification and the classification of main lineages.

Here, we present a nuclear phylogenomic tree that includes all 64 orders and 416 families of angiosperms recognized by the prevailing classification[18], using the Angiosperms353 (ref. 8) gene panel. Our sampling of 7,923 angiosperm genera (represented by 9,506 species) amounts to a 15-fold increase compared to previous work[9]. Leveraging a dataset of 200 fossil calibrations, we scale the tree to time, effectively capturing evolutionary divergences for all but the most recent 15% of angiosperm history. Although our tree broadly supports relationships predicted by previous studies primarily based on plastid data, it also shows previously unknown relationships and highlights some that remain intractable despite a vast increase in data. Gene tree conflict is tightly linked to diversification across the tree. We find evidence for high levels of conflict associated with an early burst of diversification, which is followed by an extended period of constant diversification rates underpinned by a tapestry of varied lineage-specific patterns. Diversification then increases in the Cenozoic Era, potentially driven by global climatic cooling. Our results highlight the fundamental role of botanical collections in reconstructing the tree of life to illuminate long-standing questions in angiosperm macroevolution.

## The angiosperm tree of life

Our phylogenetic tree includes 58% of the approximately 13,600 currently accepted genera of angiosperms (Fig. 1 and Supplementary Table 1; ref. 2). Together, the 7,923 genera encompass 85.7% of total known angiosperm species diversity. We produced data for 6,777 of these genera; before this study, 3,154 of these lacked publicly available genomic data, of which 393 lacked any form of DNA

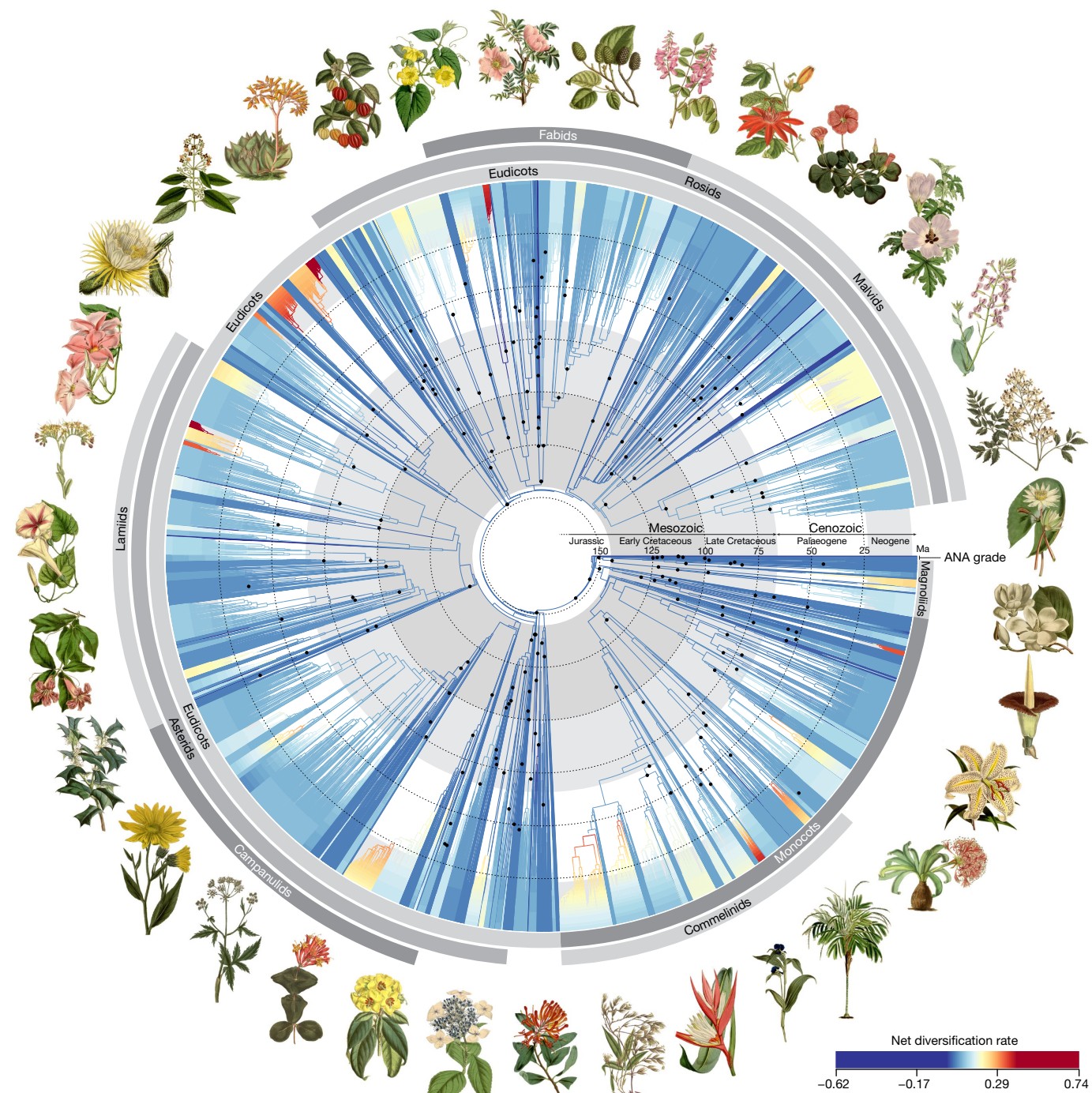

**Fig. 1 | Time-calibrated phylogenetic tree for angiosperms based on 353 nuclear genes.** All 64 orders, all 416 families and 58% (7,923) of genera are represented. The young tree is illustrated here (maximum constraint at the root node of 154 Ma), with branch colours representing net diversification rates. Black dots at nodes indicate the phylogenetic placement of fossil calibrations based on the updated AngioCal fossil calibration dataset. Note that calibrated nodes can be older than the age of the corresponding fossils owing to the use of minimum age constraints. Arcs around the tree indicate the main clades of angiosperms as circumscribed in this paper. ANA grade refers to the three consecutively diverging orders Amborellales, Nymphaeales and Austrobaileyales. Plant portraits illustrating key orders were sourced from Curtis's Botanical Magazine (Biodiversity Heritage Library). These portraits, by S. Edwards, W. H. Fitch, W. J. Hooker, J. McNab and M. Smith, were first published between 1804 and 1916 (for a key to illustrations see Supplementary Table 2). A high-resolution version of this figure can be downloaded from https://doi.org/10.5281/zenodo.10778206 (ref. 55).

sequence data. For the remaining genera, data were obtained from public repositories. Sampling for this project was possible thanks to the collaborative effort of many biodiversity institutions from around the world, including 163 herbaria in 48 countries. More than one-third of species were sourced directly from herbarium specimens, some dating back nearly 200 years. Many phylogenetically problematic lineages with unconventional genome evolution were sampled, such as holoparasites, mycoheterotrophs and aquatics. Many of the species included are threatened and four are extinct (or extinct in the wild). The resulting tree of life presented here is one of the largest genomic trees generated yet for angiosperms as a whole.

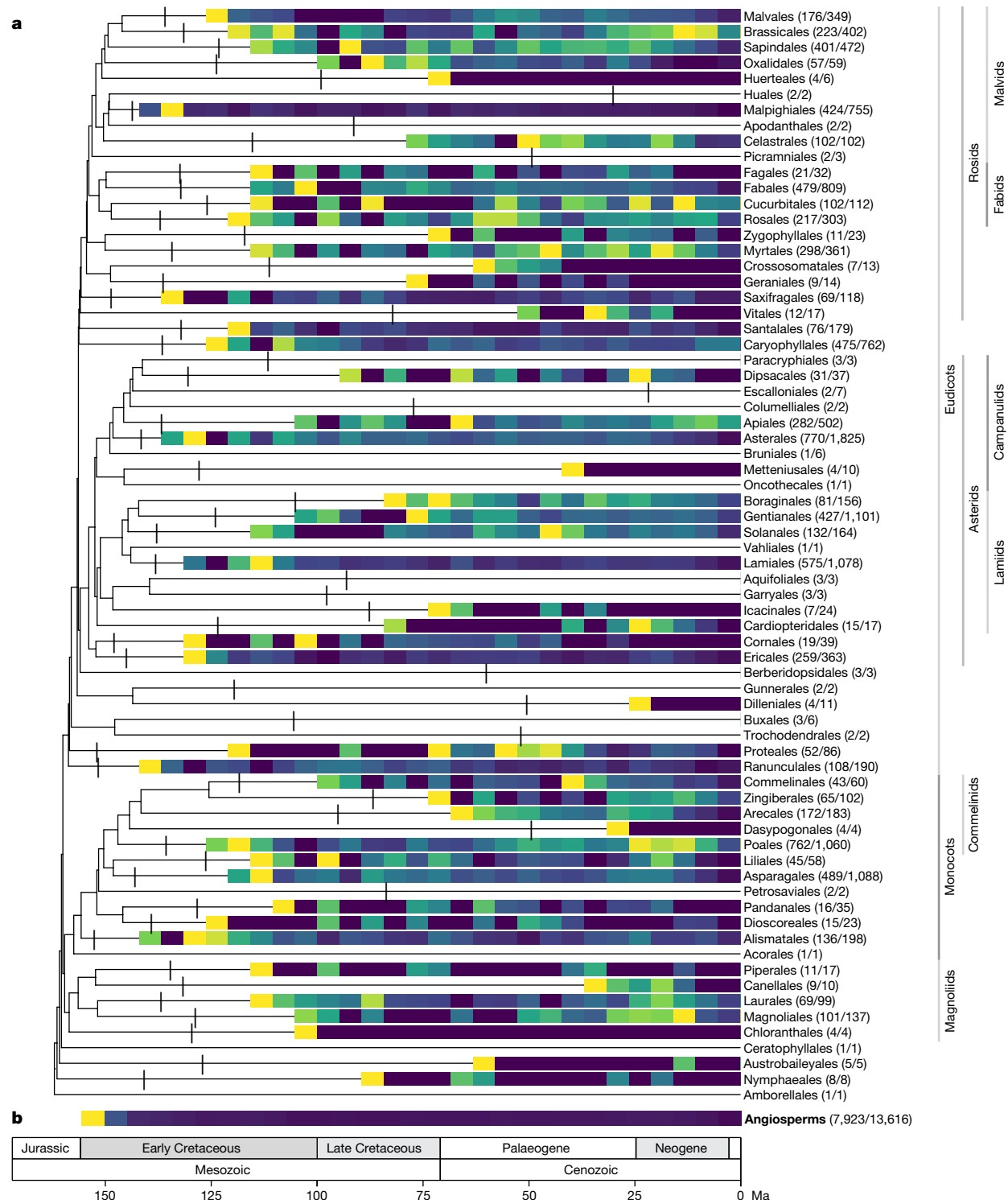

**Fig. 2 | Diversification dynamics across angiosperms.** The results illustrated are based on the young tree (maximum constraint at the root node of 154 Ma). **a**, Time-calibrated summary phylogenetic tree with LTT plots rendered as heatmaps for all orders with four or more sampled genera. The log-transformed increase in the number of lineages is depicted in 5 Myr intervals, omitting crown nodes, which disproportionately altered the visualization; crown node locations are indicated by vertical lines. The yellow to blue colour scale represents steep to shallow slopes. For each order, the numbers of sampled and total genera are provided. **b**, A global LTT heatmap for all angiosperms is shown at the bottom of the figure as a whole.

## The phylogenomic challenge

Large genomic datasets present challenges to phylogenetic inference. One issue is accurate homology assessment, which proved intractable across the full span of our dataset, even with the most advanced multiple sequence alignment methods. Another challenge is the efficient search of tree space based on gene matrices that have many more taxa than characters. We overcame both challenges with a divide-and-conquer approach (Supplementary Fig. 1). First, we computed a backbone species tree with sampling limited to five species per family (1,336 (15%) samples in total) and targeted to represent their deepest nodes (Supplementary Fig. 2). We used the

backbone species tree to delimit taxon subsets for the construction of order-level gene alignments, which were then merged into global alignments. We then computed global gene trees from the global alignments, using backbone gene trees (inferred during the estimation of the backbone species tree) as topological constraints to reduce tree space while still letting gene trees differ from each other. The smaller number of samples in the backbone dataset permits a more thorough search of tree space, resulting in greater confidence at deeper nodes than could be achieved in an unconstrained global analysis. This approach allows a trade-off between comprehensive sampling and tree search robustness while accommodating putative discordance among gene trees. Finally, we used the global gene trees to generate a global species tree in a multispecies coalescent framework (Supplementary Fig. 3).

A widespread concern in phylogenomic analysis is the presence of undetected gene copies. Our findings are unlikely to be affected by this because we used genes that have been selected to be mostly single-copy across green plants[8,9]. Although gene duplication cannot be ruled out[19], the methods we used have been shown to be robust to the presence of paralogues[20]. In addition, a full assessment of orthologues was not computationally tractable but should be undertaken when methods become available to fully unravel the complexity of genome evolution at this scale[21].

## Phylogenetic insights from nuclear data

Our results broadly corroborate the prevailing understanding of angiosperm phylogenetic relationships, which rests on three decades of molecular systematic research largely built on data from the plastid genome[3,4,18,22]. We recover all main lineages of angiosperms, namely Amborellales, Nymphaeales, Austrobaileyales, Ceratophyllales and the three larger clades, monocots, magnoliids (including Chloranthales) and eudicots (Figs. 1 and 2). Although some of the relationships among those groups, such as the placement of Amborellales as sister group to all other angiosperms, are well-established and confirmed here, others, such as the placement of Ceratophyllales, which have been unstable in previous work[4,9], remain inconclusive in our results. Despite the contrasting biological properties of the nuclear and plastid genomes (for example, size, copy number, mode of inheritance, recombination and evolutionary rate), which can lead to conflicting phylogenetic results, our findings largely support the mostly plastid-based phylogenetic classification of the Angiosperm Phylogeny Group[18] (Extended Data Fig. 1). For example, 58 of the 64 now accepted orders and 406 of the 416 families are recovered as monophyletic (excluding artefacts; Supplementary Table 1). The most striking exception is the non-monophyly of Asteraceae, the largest angiosperm family comprising the sunflowers and their relatives. Our tree also confirms 85% of the relationships among families recovered by ref. 4 using plastid genomes (Supplementary Fig. 4).

The overall stability of established relationships is unevenly distributed across the tree, as observed in contrasting patterns in the main eudicot clades, the asterids and rosids, which account for 35% and 29% of angiosperm diversity, respectively[2]. The relationships among main orders of asterids are stable[9], with a clade comprising Ericales and Cornales sister to all other asterids and the remaining 15 orders divided in two main clades (campanulids and lamiids), both long characterized by their contrasting floral ontogeny[23]. Relationships contrasting with the status quo are mostly restricted to small orders, such as the paraphyly of Aquifoliales, Bruniales and Icacinales. These DNA-defined orders were consistently recovered as highly supported clades in plastome analyses[4,24] but they lack morphological cohesion. Given their placement in our phylogenetic tree and unique morphologies, these changes, although small, will alter our understanding of the evolution of asterids.

By contrast to asterids, our findings in rosids conflict markedly with plastid-based evidence. First, we resolve Saxifragales, rather than

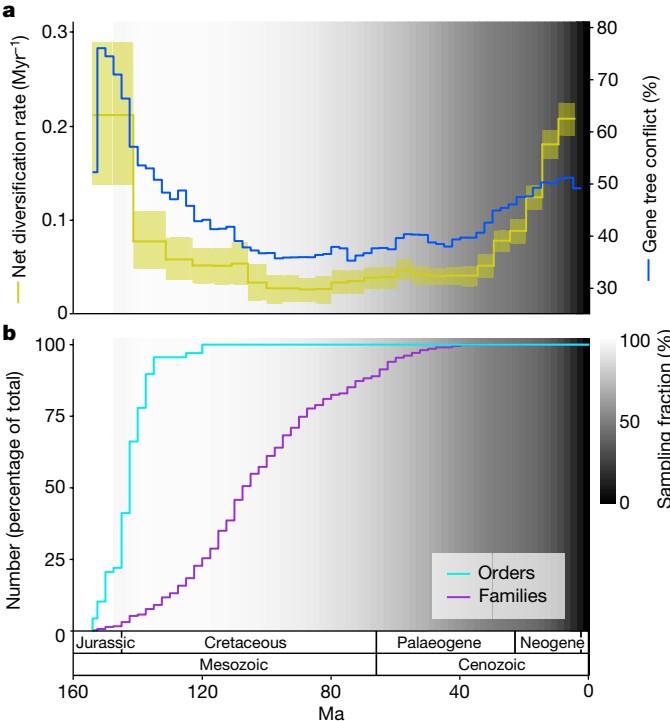

**Fig. 3 | Angiosperm-wide diversification and gene tree conflict through time.** The results illustrated are based on the young tree (maximum constraint at the root node of 154 Ma). See Extended Data Fig. 5 for results based on the old tree. **a**, Estimated net diversification rate through time (yellow, left $y$ axis) and the level of gene tree conflict through time (blue, right $y$ axis). Net diversification rates are estimated with a model that enables speciation rates to vary between time intervals; the line is the posterior mean and the yellow shaded area is the 95% highest posterior density. Gene tree conflict is calculated from the percentage of gene trees that do not share a congruent bipartition with each species tree branch, with the plotted value being the mean across all species tree branches that cross each 2.5 Myr time slice. **b**, Cumulative percentage of extant orders and families that have originated through time. In both **a** and **b**, the background grey-scale gradient is the estimated percentage of extant lineages represented in the species tree through time (sampling fraction).

Vitales[4], as sister to the remainder of rosids. In rosids, the fabid and malvid subclades, recovered as reciprocally monophyletic by plastid data[4,22], are substantially rearranged into a grade of four orders subtending two well-supported sister clades, which we designate here as the recircumscribed fabids and malvids. The new fabid clade (Cucurbitales, Fabales, Fagales and Rosales) has long been characterized by symbiotic nitrogen fixation[25]. In the new malvids (Brassicales, Celastrales, Huerteales, Malpighiales, Malvales, Oxalidales, Picramniales and Sapindales), Oxalidales is resolved as two independent lineages, the core emerging closer to Brassicales, Malvales and Sapindales, whereas Huaceae emerges in the position conventionally occupied by Oxalidales, that is, closer to Malpighiales and Celastrales (the former Celastrales–Oxalidales–Malpighiales (COM) clade[18]).

Notwithstanding the many well-supported confirmatory and new findings, some key relationships remain contentious and cannot be resolved by our data. These areas of high gene tree conflict often coincide with biological processes that confound phylogenetic inference. For example, the uncertain placements of eudicot orders Caryophyllales, Dilleniales and Gunnerales are probably impacted by key whole genome duplications[9,26]. The poor support for relationships among magnoliids, monocots, eudicots and Ceratophyllales might be explained by ancient hybridization events, such as that recently proposed for the origin of the monocots[27]. These examples highlight

the importance of areas of poor resolution as waymarkers to biological events meriting further study.

## Time frame for angiosperm macroevolution

Our tree was analysed in combination with a dataset of 200 fossil calibrations (originally described in ref. 5, with modifications) to estimate divergence times and rates of diversification. Because the age of angiosperms is uncertain[28], we dated the tree with two different maximum constraints at the angiosperm crown node (154 and 247 million years ago (Ma), termed the young tree and old tree, respectively), which reflect realistic upper and lower bounds for the maximum age of this node[5,28]. These different constraints affected age estimates across angiosperms (Extended Data Fig. 2, Supplementary Fig. 5 and Supplementary Table 3). For example, in the young tree, stem node age estimates for Nymphaeales, Austrobaileyales and Ceratophyllales were 153, 152 and 152 Ma, respectively, whereas in the old tree the equivalent age estimates were 245, 244 and 243 Ma. Likewise, for larger clades such as magnoliids, monocots and eudicots, crown node age estimates were 151, 149 and 151 Ma in the young tree and 238, 237 and 241 Ma in the old tree. This range in age estimates is consistent with the most comprehensive comparable study[5] (Extended Data Fig. 3) but our trees provide age estimates for a further 7,000 nodes. In subsequent analyses, we indicate if differing age estimates between the young tree and old tree cause substantially different interpretations of angiosperm diversification.

With our sampling across angiosperms, we ensured that deeper branching events leading to extant lineages are comprehensively represented, while recognizing that extinct lineages are inaccessible to genomic methods. However, our dated trees are sparsely sampled at the species-level, meaning that branching events are incompletely represented towards the present, limiting diversification inferences in that time window. To address this, we developed a simulation-based approach to quantify the sampling fraction through time. For both dated trees, the lineage representation begins to drop substantially (below 75%) around 50 Ma (Supplementary Fig. 6). However, the most dramatic fall in lineage representation occurs in the most recent 20 Myr, in which it falls from around 50% to slightly more than 1% at present. Our investigation of angiosperm diversification should be interpreted with this broader context in mind. In particular, inferences in the most recent 20 Myr may be updated in the future with denser species sampling.

## The diversification of angiosperms
### Diversification linked to gene conflict

We used our dated trees to reconstruct both diversification and gene tree conflict across a broad range of temporal and phylogenetic scales and investigate the relationship between them. We show that throughout angiosperm macroevolution, elevated gene tree conflict was tightly associated with elevated diversification. At a general level, this relationship is visible by simply comparing estimated diversification rates with gene tree conflict across all angiosperms through time (Fig. 3a). Meanwhile, in a branch-specific analysis using the temporal duration of branches as a proxy for the rate at which branches are diversifying, we also show that conflict and diversification rate are positively correlated (Extended Data Fig. 4) ($P < 0.001$, $r^2 = 0.51$).

To characterize the theoretical basis of this relationship, we simulated species trees with corresponding gene trees under different diversification scenarios in a multispecies coalescent framework. These simulations showed that gene tree conflict is positively correlated with diversification when caused by incomplete lineage sorting, assuming that effective population size is constant (Supplementary Fig. 7). Our empirical results are largely consistent with such a scenario. Other potential causes of gene tree conflict such as whole genome duplication

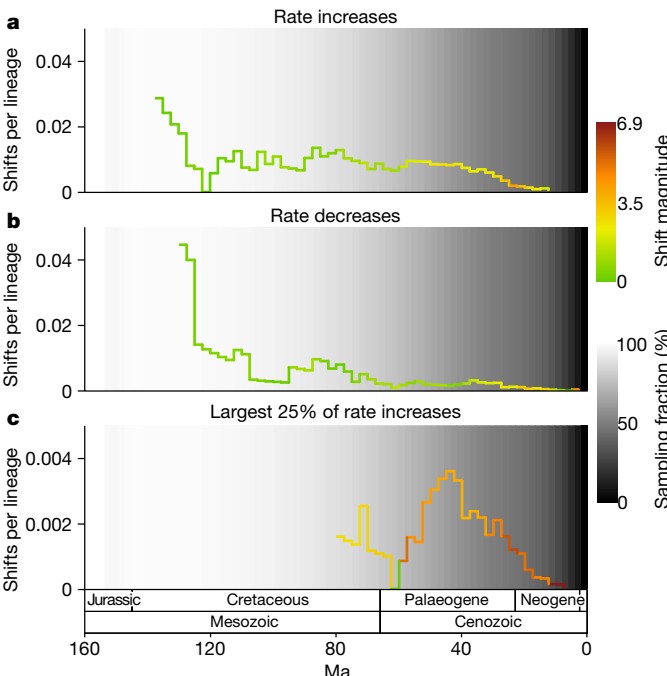

**Fig. 4 | Summary of lineage-specific diversification rate shifts estimated by BAMM.** The results illustrated are based on the young tree (maximum constraint at the root node of 154 Ma). See Extended Data Fig. 6 for results based on the old tree. **a**, Diversification rate increases per LTT. The colour corresponds to the average magnitude of the rate increases during the time period. **b**, Equivalent to **a** but for rate decreases. **c**, Equivalent to **a** but focusing on the largest 25% of diversification rate increases. In **a**, **b** and **c**, the number of shifts is from the maximum a posteriori shift configuration with the prior for the number of shifts set to 10 and the background grey-scale gradient is the estimated percentage of extant lineages represented in the species tree through time (sampling fraction).

and hybridization may also be associated with rapid diversification and have been recorded extensively throughout angiosperms[29,30]. Overall, however, gene tree conflict seems to be reliable corroborating evidence for investigating temporal patterns of angiosperm diversification.

### Early burst of angiosperm diversification

Our lineage-through-time (LTT) heatmap and diversification rate estimates through time both indicate an explosive early phase of diversification of extant lineages during the Late Jurassic and Early Cretaceous Periods (Fig. 2b and Fig. 3a). An early burst of angiosperm diversification, popularized as 'Darwin's abominable mystery'[31,32], is expected given the sudden emergence of diverse angiosperm fossils during the Early Cretaceous[11,33–35]. Phylogenetic studies based on single or few genes have also implied that angiosperms diversified rapidly in the Early Cretaceous[7,36–38]. Our dated tree corroborates the existence of a distinct early burst of diversification, associated with high levels of gene tree conflict (Fig. 3a and Supplementary Fig. 8), further increasing our confidence in this finding.

More than 80% of extant angiosperm orders originated during the early burst of diversification (Fig. 3b). Although not strictly comparable because of their subjective delimitation, orders represent the main components of angiosperm feature diversity, which have arisen rapidly after the crown node of angiosperms. In the young tree (Fig. 3), the early burst occurs during the Cretaceous, consistent with the hypothesis that a Cretaceous terrestrial revolution was triggered by the establishment of main angiosperm lineages[14,39,40]. More controversially, the old tree places the early burst in the Triassic Period (Extended Data Fig. 5), which is dramatically at variance with the palaeobotanical record[33,34],

highlighting that current molecular dating methods are unable to resolve the age of angiosperms[28].

## A tapestry of lineage-specific histories

Following the early burst, overall rates of diversification across angiosperms continued at a lower, constant pace for at least 80 Myr (Fig. 3a), during which time around three-quarters of all families originated (Fig. 3b). As expected, this phase of slower diversification was associated with lower levels of gene tree conflict. Despite the constancy of overall rates, diversification during this period was underpinned by a complex tapestry of lineage-specific patterns. This is illustrated by the LTT heatmap, which shows profound differences in diversification trajectories among orders (Fig. 2) and by the estimation of around 160 lineage-specific diversification rate shifts in angiosperms, most of which occur during this period. These rate shifts have a widespread phylogenetic distribution, with most orders containing at least one rate shift and many containing several nested shifts (Supplementary Table 4). The importance of nested rate shifts is highlighted extensively in discussions of evolutionary radiation[41,42] and underpins the continual response of diversification to dynamic extrinsic and intrinsic conditions. However, because these rate shifts are temporally scattered, as also shown by ref. 43, they do not lead to observable global rate shifts across angiosperms.

## A Cenozoic diversification surge

A second surge in angiosperm diversification occurred during the Cenozoic Era (Fig. 3a). The occurrence of this surge, despite the already high standing diversity of angiosperms at the time, suggests that diversification was unaffected by diversity-dependent processes, that is, the filling of available niche space as clades diversify[44]. Instead, this finding is consistent with previously proposed positive feedbacks between increased diversity and increased rates of diversification in angiosperms[14], alongside more positive feedbacks, for example, between angiosperm and insect diversification[45,46]. Alternatively, global climatic cooling during the Cenozoic acting as a driver of angiosperm diversification could explain this finding[7,47–49]. Importantly, an even larger Cenozoic surge would probably be inferred with increased sampling that addresses the under-representation of branching events in the recent time window. The temporal distribution of lineage-specific diversification rate shifts may offer some insight into the cause of the Cenozoic surge. Many of the largest diversification rate increases occur during the Cenozoic, whereas the number of diversification rate decreases declines markedly during this period (Fig. 4). These large rate increases may underpin the Cenozoic surge. The expansion of taxon sampling should be given priority to confirm these patterns.

## Synthesis

The nuclear phylogenomic framework presented here is the result of an ongoing initiative to complete the tree of life for all angiosperm genera[50], a milestone in our understanding of angiosperm evolutionary relationships. This study not only sheds light on much of the deep diversification history of the angiosperms but also lays foundations for future work towards a species-level tree[50]. The standardized panel of nuclear genes in our dataset paves the way for more collaborations and data integration[17,51], while the open availability of universal tools to sequence them (that is, Angiosperms353 probes[8]) has made nuclear genomic data more accessible at relatively low cost. The accelerating uptake of this approach[52–54], which is readily applicable to herbarium collections[16], indicates that large volumes of data will soon become available for a wide range of applications in plant diversity, systematic and macroevolutionary research.

Our fossil-calibrated, phylogenomic tree enables a range of unique insights into broad-scale diversification dynamics of angiosperms, substantiating the early burst of diversification anticipated by Darwin while illuminating the complexity and conflict in the lineage histories underlying it. This sets the scene for future research, extending these investigations to shallower phylogenetic scales or digging more deeply into the data to discover the processes driving angiosperm diversification, such as genomic conflict, polyploidy, selection, trait evolution and adaptation. The challenges brought by the scale of this dataset and its ongoing expansion may also catalyse the development of methods which take full advantage of the global proliferation of genomic data.

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

Alexandre R. Zuntini[1,138], Tom Carruthers[1,138], Olivier Maurin[1], Paul C. Bailey[1], Kevin Leempoel[1], Grace E. Brewer[1], Niroshini Epitawalage[1], Elaine Françoso[1,2], Berta Gallego-Paramo[1], Catherine McGinnie[1], Raquel Negrão[1], Shyamali R. Roy[1], Lalita Simpson[3], Eduardo Toledo Romero[1], Vanessa M. A. Barber[1], Laura Botigué[4], James J. Clarkson[1], Robyn S. Cowan[1], Steven Dodsworth[5], Matthew G. Johnson[6], Jan T. Kim[7], Lisa Pokorny[1,8], Norman J. Wickett[9], Guilherme M. Antar[10,11], Lucinda DeBolt[12], Karime Gutierrez[12], Kasper P. Hendriks[13,14], Alina Hoewener[15], Ai-Qun Hu[1], Elizabeth M. Joyce[3,16], Izai A. B. S. Kikuchi[17], Isabel Larridon[1], Drew A. Larson[18], Elton John de Lírio[10], Jing-Xia Liu[19], Panagiota Malakasi[1], Natalia A. S. Przelomska[1,5],

Toral Shah[1], Juan Viruel[1], Theodore R. Allnutt[20], Gabriel K. Ameka[21], Rose L. Andrew[22], Marc S. Appelhans[23], Montserrat Arista[24], María Jesús Ariza[25], Juan Arroyo[24], Watchara Arthan[1], Julien B. Bachelier[26], C. Donovan Bailey[27], Helen F. Barnes[20], Matthew D. Barrett[3], Russell L. Barrett[28], Randall J. Bayer[29], Michael J. Bayly[30], Ed Biffin[31], Nicky Biggs[1], Joanne L. Birch[30], Diego Bogarín[14,32], Renata Borosova[1], Alexander M. C. Bowles[33], Peter C. Boyce[34], Gemma L. C. Bramley[1], Marie Briggs[1], Linda Broadhurst[35], Gillian K. Brown[36], Jeremy J. Bruhl[22], Anne Bruneau[37], Sven Buerki[38], Edie Burns[1], Margaret Byrne[39], Stuart Cable[1], Ainsley Calladine[31], Martin W. Callmander[40], Ángela Cano[41], David J. Cantrill[20], Warren M. Cardinal-McTeague[42], Mónica M. Carlsen[43], Abigail J. A. Carruthers[1], Alejandra de Castro Mateo[24], Mark W. Chase[1,44], Lars W. Chatrou[45], Martin Cheek[1], Shilin Chen[46,47], Maarten J. M. Christenhusz[1,48,49], Pascal-Antoine Christin[50], Mark A. Clements[35], Skye C. Coffey[51], John G. Conran[52], Xavier Cornejo[53], Thomas L. P. Couvreur[54], Ian D. Cowie[55], Laszlo Csiba[1], Iain Darbyshire[1], Gerrit Davidse[43], Nina M. J. Davies[1], Aaron P. Davis[1], Kor-jent van Dijk[56], Stephen R. Downie[57], Marco F. Duretto[28], Melvin R. Duvall[58], Sara L. Edwards[1], Urs Eggli[59], Roy H. J. Erkens[14,60,61], Marcial Escudero[24], Manuel de la Estrella[62], Federico Fabriani[45], Michael F. Fay[1], Paola de L. Ferreira[63,64], Sarah Z. Ficinski[1], Rachael M. Fowler[30], Sue Frisby[1], Lin Fu[65], Tim Fulcher[1], Mercè Galbany-Casals[66], Elliot M. Gardner[67], Dmitry A. German[68], Augusto Giaretta[69], Marc Gibernau[70], Lynn J. Gillespie[71], Cynthia C. González[72], David J. Goyder[1], Sean W. Graham[17], Aurélie Grall[1], Laura Green[1], Bee F. Gunn[20], Diego G. Gutiérrez[73], Jan Hackel[1,74], Thomas Haevermans[75], Anna Haigh[1], Jocelyn C. Hall[76], Tony Hall[1], Melissa A. Harrison[3], Sebastian A. Hatt[1], Oriane Hidalgo[77], Trevor R. Hodkinson[78], Gareth D. Holmes[20], Helen C. F. Hopkins[1], Christopher J. Jackson[20], Shelley A. James[51], Richard W. Jobson[28], Gudrun Kadereit[79], Imalka M. Kahandawala[1], Kent Kainulainen[80], Masahiro Kato[81], Elizabeth A. Kellogg[43], Graham J. King[83], Beata Klejevskaja[44], Bente B. Klitgaard[1], Ronell R. Klopper[85,86], Sandra Knapp[87], Marcus A. Koch[88], James H. Leebens-Mack[89], Frederic Lens[14], Christine J. Leon[1], Étienne Léveillé-Bourret[90], Gwilym P. Lewis[1], De-Zhu Li[91], Lan Li[91], Sigrid Liede-Schumann[92], Tatyana Livshultz[93,94], David Lorence[95], Meng Lu[1], Patricia Lu-Irving[28], Jaquelini Luber[96], Eve J. Lucas[1], Manuel Luján[97], Mabel Lum[97], Terry D. Macfarlane[51], Carlos Magdalena[1], Vidal F. Mansano[96], Lizo E. Masters[1], Simon J. Mayo[1], Kristina McColl[28], Angela J. McDonnell[98], Andrew E. McDougall[56], Todd G. B. McLay[20], Hannah McPherson[28], Rosa I. Meneses[99], Vincent S. F. T. Merckx[14], Fabián A. Michelangeli[100], John D. Mitchell[100], Alexandre K. Monro[1], Michael J. Moore[101], Taryn L. Mueller[102], Klaus Mummenhoff[13], Jérôme Munzinger[103], Priscilla Muriel[104], Daniel J. Murphy[20], Katharina Nargar[3,35], Lars Nauheimer[3], Francis J. Nge[31], Reto Nyffeler[105], Andrés Orejuela[106,107], Edgardo M. Ortiz[15], Luis Palazzesi[73], Ariane Luna Peixoto[96], Susan K. Pell[108], Jaume Pellicer[77], Darin S. Penneys[109], Oscar A. Perez-Escobar[1], Claes Persson[110], Marc Pignal[75], Yohan Pillon[111], José R. Pirani[10], Gregory M. Plunkett[100], Robyn F. Powell[1], Ghillean T. Prance[1], Carmen Puglisi[1,43], Ming Qin[65], Richard K. Rabeler[18], Paul E. J. Rees[1], Matthew Renner[28], Eric H. Roalson[112], Michele Rodda[113], Zachary S. Rogers[114], Saba Rokni[1], Rolf Rutishauser[105], Miguel F. de Salas[115], Hanno Schaefer[15], Rowan J. Schley[116], Alexander Schmidt-Lebuhn[35], Alison Shapcott[117], Ihsan Al-Shehbaz[43], Kelly A. Shepherd[51], Mark P. Simmons[118], André O. Simões[119], Ana Rita G. Simões[1], Michelle Siros[1,120], Eric C. Smidt[121], James F. Smith[38], Neil Snow[122], Douglas E. Soltis[123], Pamela S. Soltis[123], Robert J. Soreng[124], Cynthia A. Sothers[1], Julian R. Starr[125], Peter F. Stevens[43], Shannon C. K. Straub[126], Lena Struwe[127], Jennifer M. Taylor[91], Ian R. H. Telford[22], Andrew J. Thornhill[22,31,52], Ifeanna Tooth[28], Anna Trias-Blasi[1], Frank Udovicic[20], Timothy M. A. Utteridge[1], Jose C. Del Valle[24], G. Anthony Verboom[128], Helen P. Vonow[31], Maria S. Vorontsova[1], Jurriaan M. de Vos[129], Noor Al-Wattar[1], Michelle Waycott[31,52], Cassiano A. D. Welker[130], Adam J. White[131], Jan J. Wieringa[14], Luis T. Williamson[56], Trevor C. Wilson[28], Sin Yeng Wong[132], Lisa A. Woods[28], Roseina Woods[1], Stuart Worboys[3], Martin Xanthos[1], Ya Yang[133], Yu-Xiao Zhang[134], Meng-Yuan Zhou[19], Sue Zmarzty[1], Fernando O. Zuloaga[135], Alexandre Antonelli[1,110,136,137], Sidonie Bellot[1], Darren M. Crayn[3], Olwen M. Grace[1,106], Paul J. Kersey[1], Ilia J. Leitch[1], Hervé Sauquet[28], Stephen A. Smith[18,139], Wolf L. Eiserhardt[1,64,139], Félix Forest[1,139] & William J. Baker[1,64,139] ✉

[1]Royal Botanic Gardens, Kew, Richmond, UK. [2]Centre for Ecology, Evolution and Behaviour, Department of Biological Sciences, School of Life Sciences and the Environment, Royal Holloway University of London, London, UK. [3]Australian Tropical Herbarium, James Cook University, Smithfield, Queensland, Australia. [4]Centre for Research in Agricultural Genomics (CRAG), CSIC-IRTA-UAB-UB, Campus UAB, Barcelona, Spain. [5]School of Biological Sciences, University of Portsmouth, Portsmouth, UK. [6]Texas Tech University, Lubbock, TX, USA. [7]School of Physics, Engineering and Computer Science, University of Hertfordshire, Hatfield, UK. [8]Department of Biodiversity and Conservation, Real Jardín Botánico (RJB-CSIC), Madrid, Spain. [9]Department of Biological Sciences, Clemson University, Clemson, SC, USA. [10]Departamento de Botânica, Instituto de Biociências, Universidade de São Paulo, São Paulo, Brazil. [11]Departamento de Ciências Agrárias e Biológicas, Centro Universitário Norte do Espírito Santo, Universidade Federal do Espírito Santo, São Mateus, Brazil. [12]Smith College, Northampton, MA, USA. [13]Department of Biology, University of Osnabrück, Osnabrück, Germany. [14]Naturalis Biodiversity Center, Leiden, The Netherlands. [15]Plant Biodiversity, Technical University Munich, Freising, Germany. [16]Systematic, Biodiversity and Evolution of Plants, Ludwig Maximilian University of Munich, Munich, Germany. [17]Department of Botany, University of British Columbia, Vancouver, British Columbia, Canada. [18]Department of Ecology & Evolutionary Biology, University of Michigan, Ann Arbor, MI, USA. [19]Germplasm Bank of Wild Species, Kunming Institute of Botany, Chinese Academy of Sciences, Kunming, China. [20]Royal Botanic Gardens Victoria, Melbourne, Victoria, Australia. [21]Department of Plant and Environmental Biology, University of Ghana, Accra, Ghana. [22]Botany and N.C.W. Beadle Herbarium, University of New England, Armidale, New South Wales, Australia. [23]Department of Systematics, Biodiversity and Evolution of Plants, Albrecht-von-Haller Institute of Plant Sciences, University of Göttingen, Göttingen, Germany. [24]Departamento de Biología Vegetal y Ecología, Facultad de Biología, Universidad de Sevilla, Seville, Spain. [25]General Research Services, Herbario SEV, CITIUS, Universidad de Sevilla, Seville, Spain. [26]Institute of Biology,

Freie Universität, Berlin, Germany. [27]Department of Biology, New Mexico State University, Las Cruces, NM, USA. [28]National Herbarium of NSW, Botanic Gardens of Sydney, Mount Annan, New South Wales, Australia. [29]Department of Biological Sciences, University of Memphis, Memphis, TN, USA. [30]School of BioSciences, The University of Melbourne, Parkville, Victoria, Australia. [31]State Herbarium of South Australia, Botanic Gardens and State Herbarium, Adelaide, South Australia, Australia. [32]Jardín Botánico Lankester, Universidad de Costa Rica, Cartago, Costa Rica. [33]School of Geographical Sciences, University of Bristol, Bristol, UK. [34]Centro Studi Erbario Tropicale, Dipartimento di Biologia, University of Florence, Florence, Italy. [35]Centre for Australian National Biodiversity Research, National Research Collections Australia, CSIRO, Canberra, Australian Capital Territory, Australia. [36]Queensland Herbarium and Biodiversity Science, Brisbane Botanic Gardens, Toowong, Queensland, Australia. [37]Institut de Recherche en Biologie Végétale and Département de Sciences Biologiques, University of Montreal, Montreal, Quebec, Canada. [38]Department of Biological Sciences, Boise State University, Boise, ID, USA. [39]Biodiversity and Conservation Science, Department of Biodiversity, Conservation and Attractions, Government of Western Australia, Kensington, Western Australia, Australia. [40]Conservatoire et Jardin Botaniques de Genève, Chambésy, Switzerland. [41]Cambridge University Botanic Garden, Cambridge, UK. [42]Department of Forest and Conservation Sciences, University of British Columbia, Vancouver, British Columbia, Canada. [43]Missouri Botanical Garden, St. Louis, MO, USA. [44]Department of Environment and Agriculture, Curtin University, Bentley, Western Australia, Australia. [45]Department of Biology, Ghent University, Ghent, Belgium. [46]Institute of Herbgenomics, Chengdu University of Traditional Chinese Medicine, Chengdu, China. [47]Institute of Medicinal Plant Development, Chinese Academy of Medical Sciences, Beijing, China. [48]Department of Environment and Agriculture, Curtin University, Perth, Western Australia, Australia. [49]Plant Gateway, Den Haag, The Netherlands. [50]Ecology and Evolutionary Biology, School of Biosciences, University of Sheffield, Sheffield, UK. [51]Western Australian Herbarium, Department of Biodiversity, Conservation and Attractions, Government of Western Australia, Kensington, Western Australia, Australia. [52]School of Biological Sciences, The University of Adelaide, Adelaide, South Australia, Australia. [53]Herbario GUAY, Facultad de Ciencias Naturales, Universidad de Guayaquil, Guayaquil, Ecuador. [54]DIADE, Université Montpellier, CIRAD IRD, Montpellier, France. [55]Northern Territory Herbarium Department of Environment Parks & Water Security, Northern Territory Government, Palmerston, Northern Territory, Australia. [56]The University of Adelaide, North Terrace Campus, Adelaide, South Australia, Australia. [57]Department of Plant Biology, University of Illinois at Urbana-Champaign, Urbana, IL, USA. [58]Department of Biological Sciences and Institute for the Study of the Environment, Sustainability and Energy, Northern Illinois University, DeKalb, IL, USA. [59]Sukkulenten-Sammlung Zürich/ Grün Stadt Zürich, Zürich, Switzerland. [60]Maastricht Science Programme, Maastricht University, Maastricht, The Netherlands. [61]System Earth Science, Maastricht University, Venlo, The Netherlands. [62]Departamento de Botánica, Ecología y Fisiología Vegetal, Facultad de Ciencias, Universidad de Córdoba, Córdoba, Spain. [63]Departamento de Biologia, Faculdade de Ciências e Letras de Ribeirão Preto, Universidade de São Paulo, São Paulo, Brazil. [64]Department of Biology, Aarhus University, Aarhus, Denmark. [65]South China Botanical Garden, Chinese Academy of Sciences, Guangzhou, China. [66]Systematics and Evolution of Vascular Plants (UAB)—Associated Unit to CSIC by IBB, Departament de Biologia Animal, Biologia Vegetal i Ecologia, Facultat de Biociències, Universitat Autònoma de Barcelona, Bellaterra, Spain. [67]Department of Biology, Case Western Reserve University, Cleveland, OH, USA. [68]Altai State University, Barnaul, Russia. [69]Faculdade de Ciências Biológicas e Ambientais, Universidade Federal da Grande Dourados, Dourados, Brazil. [70]Laboratoire Sciences Pour l'Environnement, Université de Corse, Ajaccio, France. [71]Canadian Museum of Nature, Ottawa, Ontario, Canada. [72]Herbario Trelew, Universidad Nacional de la Patagonia San Juan Bosco, Trelew, Argentina. [73]Museo Argentino de Ciencias Naturales (MACN-CONICET), Buenos Aires, Argentina. [74]Department of Biology, Universität Marburg, Marburg, Germany. [75]Institut de Systématique, Evolution, Biodiversité, Muséum National d'Histoire Naturelle, Paris, France. [76]Department of Biological Sciences, University of Alberta, Edmonton, Alberta, Canada. [77]Institut Botànic de Barcelona (IBB CSIC-Ajuntament de Barcelona), Barcelona, Spain. [78]Botany, School of Natural Sciences, Trinity College Dublin, The University of Dublin, Dublin, Ireland. [79]Prinzessin Therese von Bayern-Lehrstuhl für Systematik, Biodiversität & Evolution der Pflanzen, Ludwig-Maximilians-Universität München, Botanische Staatssammlung München, Botanischer Garten München-Nymphenburg, Munich, Germany. [80]Gothenburg Botanical Garden, Gothenburg, Sweden. [81]National Museum of Nature and Science, Tsukuba, Japan. [82]Donald Danforth Plant Science Center, St. Louis, MO, USA. [83]Southern Cross University, Lismore, New South Wales, Australia. [84]Synergy SRG, Luton, UK. [85]Foundational Biodiversity Science Division, South African National Biodiversity Institute, Pretoria, South Africa. [86]Department of Plant and Soil Sciences, University of Pretoria, Pretoria, South Africa. [87]Natural History Museum, London, UK. [88]Centre for Organismal Studies, Biodiversity and Plant Systematics, Heidelberg University, Heidelberg, Germany. [89]Department of Plant Biology, University of Georgia, Athens, GA, USA. [90]Institut de Recherche en Biologie Végétale, University of Montreal, Montreal, Quebec, Canada. [91]CSIRO, Canberra, Australian Capital Territory, Australia. [92]Department of Plant Systematics, University of Bayreuth, Bayreuth, Germany. [93]Department of Biodiversity, Earth and Environmental Sciences, Drexel University, Philadelphia, PA, USA. [94]Academy of Natural Science, Drexel University, Philadelphia, PA, USA. [95]National Tropical Botanical Garden, Kalaheo, HI, USA. [96]Instituto de Pesquisas Jardim Botânico do Rio de Janeiro, Rio de Janeiro, Brazil. [97]Bioplatforms Australia Ltd, Sydney, New South Wales, Australia. [98]Department of Biological Sciences, Saint Cloud State University, Saint Cloud, MN, USA. [99]Instituto de Arqueología y Antropología, Universidad Católica del Norte, San Pedro de Atacama, Chile. [100]New York Botanical Garden, Bronx, NY, USA. [101]Department of Biology, Oberlin College, Oberlin, OH, USA. [102]Department of Ecology, Evolution & Behavior, University of Minnesota, St. Paul, MN, USA. [103]AMAP Lab, Université Montpellier, IRD, CIRAD, CNRS INRAE, Montpellier, France. [104]Laboratorio de Ecofisiología, Escuela de Ciencias Biológicas, Pontificia Universidad Católica del Ecuador, Quito, Ecuador. [105]Department of Systematic and Evolutionary Botany, University of Zürich, Zürich, Switzerland. [106]Royal Botanic Garden Edinburgh, Edinburgh, UK. [107]Grupo de Investigación en Recursos Naturales Amazónicos, Instituto Tecnológico del Putumayo, Mocoa, Colombia. [108]US Botanic Garden, Washington, DC, USA. [109]Department of Biology and Marine Biology, University of North Carolina Wilmington, Wilmington, NC, USA. [110]Department of Biological and Environmental Sciences, University of Gothenburg, Gothenburg, Sweden. [111]LSTM Université Montpellier, CIRADIRD, Montpellier, France. [112]School of Biological Sciences, Washington State University, Pullman, WA, USA. [113]National Parks Board, Singapore Botanic Gardens, Singapore, Singapore. [114]New Mexico State University, Las Cruces, NM, USA. [115]Tasmanian Herbarium, University of Tasmania, Sandy Bay, Tasmania, Australia. [116]University of Exeter, Exeter, UK. [117]School of Science Technology and Engineering, Center for Bioinnovation, University Sunshine Coast, Sippy Downs, Queensland, Australia. [118]Department of Biology, Colorado State University, Fort Collins, CO, USA. [119]Departamento de Biologia Vegetal, Universidade Estadual de Campinas, Campinas, Brazil. [120]University of California, San Francisco, San Francisco, CA, USA. [121]Departamento de Botânica, Universidade Federal do Paraná, Curitiba, Brazil. [122]Pittsburg State University, Pittsburg, KS, USA. [123]Florida Museum of Natural History, University of Florida, Gainesville, FL, USA. [124]Smithsonian Institution, Washington, DC, USA. [125]Department of Biology, University of Ottawa, Ottawa, Ontario, Canada. [126]Hobart and William Smith Colleges, Geneva, NY, USA. [127]Rutgers University, New Brunswick, NJ, USA. [128]Department of Biological Sciences and Bolus Herbarium, University of Cape Town, Cape Town, South Africa. [129]Department of Environmental Sciences—Botany, University of Basel, Basel, Switzerland. [130]Instituto de Biologia, Universidade Federal de Uberlândia, Uberlândia, Brazil. [131]Australian National Herbarium, Centre for Australian National Biodiversity Research, National Research Collections Australia, CSIRO, Canberra, Australian Capital Territory, Australia. [132]Institute of Biodiversity And Environmental Conservation, Universiti Malaysia Sarawak, Samarahan, Malaysia. [133]University of Minnesota-Twin Cities, St. Paul, MN, USA. [134]Southwest Forestry University, Kunming, China. [135]Instituto de Botánica Darwinion, San Isidro, Argentina. [136]Gothenburg Global Biodiversity Centre, University of Gothenburg, Gothenburg, Sweden. [137]Department of Biology, University of Oxford, Oxford, UK. [138]These authors contributed equally: Alexandre R. Zuntini, Tom Carruthers. [139]These authors jointly supervised this work: Stephen A. Smith, Wolf L. Eiserhardt, Félix Forest, William J. Baker. ✉e-mail: w.baker@kew.org

# Methods

As part of the Plant and Fungal Trees of Life (PAFTOL) Project at the Royal Botanic Gardens, Kew[50], we assembled a nuclear genomic dataset consisting of newly generated data and data mined from public repositories. Our objective was to sample at least 50% of all angiosperm genera, with genera selected in a phylogenetically representative manner on the basis of published research. To avoid excessive imbalance in the tree, we included only one sample per species and a maximum of three species per genus. When several samples were available for the same species, we selected those with the largest amount of data, that is, more genes and a higher sum of gene length. For genera with several species available, the criterion for selection was primarily phylogenetic representation followed by amount of data. One species of each gymnosperm family was selected to form the outgroup, totalling 12 samples.

We produced target sequence capture data for 7,561 samples using the universal Angiosperms353 probe set[8] following established laboratory protocols[50,56]. We complemented our dataset with publicly available data for 2,054 species, sourced from the One Thousand Plant Transcriptomes Initiative[9] (OneKP; 564 samples), annotated and unannotated genomes (151 samples) and the sequence read archive (SRA; 1,339 samples), the last including transcriptomes (for example, see refs. 57,58) and target capture data (for example, see refs. 59,60). To standardize taxonomy and nomenclature, all species names and families were harmonized with the World Checklist of Vascular Plants[2] and orders with APG IV if possible[18].

## Sequence recovery

Sequence recovery was carried out in two ways, depending on the type of input data. For recovery on the basis of raw reads, that is, Angiosperms353 data or data mined from the SRA, we used HybPiper v.1.31 (ref. 61), embedded in a bespoke pipeline (https://github.com/baileyp1/PhylogenomicsPipelines). Raw reads were trimmed using Trimmomatic[62] to remove low-quality bases and short sequences. In HybPiper, reads were initially binned into genes using BLASTN and an amino acid target file as reference (Supplementary File 1). Individual genes were assembled de novo using SPADES[63] and refined by joining and trimming gene contigs to match coding regions using Exonerate[64]. For genes with paralogue warnings, only the putative orthologue as identified by HybPiper was used. Exclusion of genes with several copies per species has been shown to have negligible impact on species tree inference when it is performed under a multispecies coalescent framework, as described below[20]. Conversely, the inclusion of several copies per species would have rendered our study computationally intractable. Gene sequences from assembled genomes and OneKP transcriptomes were recovered using custom scripts described in ref. 50. Briefly, the assembled sequences were searched against the target file mentioned above using BLASTN, selecting the best match for each gene and trimming it to the BLAST hit. For a few Angiosperms353 samples that represented the sole accession of their respective families (*Ixonanthes reticulata*, *Mitrastemon matudae* and *Tetracarpaea tasmannica*) and had poor recovery from HybPiper (that is, below 5 kilobase pairs (kb) in total sum of contig length), recovery was undertaken following ref. 50, using less stringent recovery thresholds. The average recovery per order is presented in Supplementary Fig. 9.

## Phylogenetic inference

To analyse the dataset, we devised a divide-and-conquer approach. First, we computed a backbone tree, sampling up to five species per family, to test the monophyly of orders and to rigorously explore deep relationships. We used the backbone tree to identify groups (orders or groups of orders) for multiple sequence alignment, with the aim of producing refined subalignments among closely related taxa. Subsequently, the subalignments were merged into global gene alignments and global gene trees were inferred from these using the respective gene trees from the backbone analysis as constraints. Finally, we inferred a multispecies coalescent tree using the estimated gene trees. The inference pipeline is summarized in Supplementary Fig. 1.

**Backbone tree inference.** The samples for the backbone were selected so as to represent the crown node and deepest divergences in each family. For families with five or fewer samples (279 families), all samples were included. For those with more than five samples (156), we selected the best sample (most genes and longest sequence) of each consecutively diverging clade (based on published phylogenetic evidence and preliminary analyses of our own data), until five samples were included. To evaluate the extent to which sample selection might affect the backbone tree topology, we inferred 20 backbone replicates, randomly selecting five samples for each family with more than five samples (among the 50% best samples in terms of gene number and gene length recovered). We then summarized the trees to family level and computed Robinson–Foulds distances between the backbone and the 20 replicates (Supplementary Fig. 10).

The phylogenetic reconstruction of the backbone involved up to two iterations of gene alignment and gene tree estimation, with an intermediate step of outlier removal. This was followed by species tree inference in a multispecies coalescent framework. In the first iteration, all sequences for a given gene were aligned using MAFFT v.7.480 (ref. 65) (with ffnsi method, that is, --retree 2 --maxiterate 1000) and with the direction of the sequence adjusted (--adjustdirection). After removing sites with more than 90% missing data with Phyutility[66], gene trees were estimated using IQ-TREE v.2.2.0-beta[67], keeping identical sequences in the analysis (--keep-ident), setting the substitution model to GTR + G and estimating branch support with 1,000 ultrafast bootstrap replicates[68]. Before the second iteration, we identified long branch outliers using TreeShrink[69] in 'all-genes' mode and rerooting at the centroids of the trees. A second iteration of gene alignment, removal of gappy sites and gene tree estimation was performed for genes with outliers after the removal of outlier sequences. Subsequently, the resulting gene trees were summarized into a species tree using ASTRAL III v.5.7.3, a quartet-based species tree estimation method statistically consistent with the multispecies coalescent model[70], enabling the full annotation option (-t 2), having first collapsed poorly supported nodes (ultrafast bootstrap ≤ 30%) in the input gene trees using Newick utilities[71].

**Order-level subalignments.** For the order-level subalignments, most orders were analysed individually, following the same method described for the backbone. In some cases, smaller orders (fewer than 50 samples) were analysed together with larger ones if they formed monophyletic groups in the backbone. These groups are: (1) Commelinales with Zingiberales, (2) Dioscoreales with Pandanales, (3) Fagales with Fabales, (4) Columelliales, Dipsacales, Escalloniales and Paracryphiales with Apiales, (5) all magnoliids (Canellales, Laurales, Magnoliales and Piperales) and (6) all gymnosperms together (Cycadales, Ephedrales, Gnetales, Ginkgoales and Pinales). Conversely, orders emerging as non-monophyletic in the backbone were split into monophyletic subgroups as follows: (1) Cardiopteridaceae and Stemonuraceae separate from the rest of Aquifoliales, (2) Dasypogonaceae separate from the rest of Arecales, (3) Collumelliaceae separate from the rest of Bruniales, (4) Oncothecaceae separate from the rest of Icacinales and (5) Huaceae separate from the rest of Oxalidales. The groupings of samples used in the order-level subalignments are provided in Supplementary Table 1. Very small groups, comprising one or two samples (termed orphan sequences), were not included in subalignments and were incorporated directly in global analyses.

**Global gene alignments and trees.** We produced global gene alignments by merging the order-level subalignments (before removal of gappy sites) and adding the orphan non-aligned sequences using MAFFT[65], with up to 100 refinement iterations. This approach yields

alignment across the order-level subalignments without disrupting the structure in the subalignments. The final gene alignments were cleaned by removing gappy sites. A summary of the alignments was produced with AMAS[72] (Supplementary Table 5) and the average occupancy per gene per order is presented in Supplementary Fig. 11.

We then estimated gene trees in Fasttree v.2.1.10 (ref. 73), setting the model to GTR + G, using pseudocounts to avoid biases from fragmentary sequences and increasing search thoroughness (-spr 4, -mlacc 2 and -slownni). We used the gene trees from the backbone analysis to constrain the topology of each respective global gene tree. To avoid propagating error from the backbone analysis to the global analysis, we removed potentially misleading signal from the backbone gene trees before applying them as constraints. First, branches with bootstrap values below 80% were collapsed to avoid enforcing poorly supported relationships. Second, tips placed far from the rest of their order were algorithmically removed (but retained in global gene alignments). Once global gene trees were estimated, outlier long branches were removed using TreeShrink and the set of pruned gene trees was used to compute the global species tree using ASTRAL-MP v.5.15.5 (ref. 74), after collapsing branches with poor support (that is, those with support lower than 10% in the Shimodaira–Hasegawa test).

## Divergence time estimation

Divergence times were estimated by penalized likelihood in treePL[75,76]. This method is computationally efficient for datasets of this scale and typically estimates similar divergence times to more computationally intensive Bayesian analyses. The coalescent species tree topology was used as the input tree with molecular branch lengths estimated in IQ-TREE, on the basis of a concatenated alignment of the top 25 genes selected by SortaDate[77]. Genes were selected by ranking their corresponding gene trees according to the number of congruent bipartitions with the species tree. We selected genes on this basis because high gene tree conflict leads to error in divergence time estimates[78,79].

Fossil calibrations were based on the AngioCal fossil calibration dataset described in ref. 5. We used an updated version of this dataset, referred to as AngioCal v.1.1 (Supplementary Table 6 and Supplementary File 2). Assigning fossil calibrations in this dataset to our tree topology led to 200 unique minimum age calibrations at internal nodes (Supplementary Table 7 and Supplementary Fig. 12). A maximum constraint of 154 or 247 Ma was used at the angiosperm crown node. These two values, respectively, represent a young and old constraint for the maximum age of the angiosperm crown node[5,28]. Both values are nonetheless considerably older than the oldest known crown group angiosperm fossils of around 127.2 Ma (ref. 80). Both maximum constraints, in combination with all the minimum age constraints, were used to time-calibrate the species tree. Depending on the maximum constraint at the root node, these dated phylogenetic trees are referred to as young tree and old tree, respectively. For both the young tree and old tree, four analyses were performed in treePL, using smoothing values of 0.1, 1, 10 or 100. These different smoothing values assume high to low levels of among-branch substitution rate variation.

## Sampling extant lineages through time

At 1 Myr intervals from the root age of the dated phylogenetic trees to the present, we calculated how many angiosperm lineages would have been present in a hypothetical tree that sampled 100% of extant angiosperm species diversity. We used this to quantify the proportion of extant lineages incorporated by our phylogenetic trees through time (Supplementary Methods). To do this we simulated unsampled diversity on the dated trees: the species diversity of unsampled genera was simulated as a constant-rate birth–death branching process originating in the crown group of its respective family, whilst unsampled species diversity in sampled genera was simulated as a constant-rate birth–death branching process originating at the stem node of the relevant genus. The extant diversity of each simulated branching process was determined using

the World Checklist of Vascular Plants[2]. At each time interval, we then calculated the proportional difference between the number of lineages in our dated phylogenetic tree and the hypothetical fully sampled tree.

## Diversification rate estimation

Dated trees estimated with alternative smoothing values were very similar (Extended Data Fig. 2 and Supplementary Fig. 5), so diversification rate estimates were only performed with the dated trees estimated with a smoothing value of 10. By contrast, age estimates in the young and old trees differed markedly. Diversification rate estimates were therefore performed for both these dated trees. In each case, the dated trees were pruned such that there was a maximum of one tip for each genus.

An initial analysis of diversification rates was performed by generating LTT plots as heatmaps for angiosperms as a whole, as well as for each order, with colours representing the steepness of each LTT curve at 5 Myr intervals. To calculate the steepness of the curve, we calculated the running difference between logarithmic corrected cumulative sums of lineages and applied Tukey's running median smoothing to avoid excessive noise. For order plots, the cumulative sum starts at the first branching point, that is, order crown nodes.

Time-dependent diversification parameters (speciation and extinction rates) were also explicitly estimated across all angiosperms. These analyses were performed in RevBayes with the dnEpisodicBirthDeath function[81]. The smallest time windows in which rates were estimated were 5 Ma. However, larger windows were used toward the root of the tree such that there were at least 50 branching events in each time window. Three different models were used: equal rates of speciation and extinction across all windows; variable rates of speciation across windows but equal rates of extinction; and equal rates of speciation across windows but variable rates of extinction. Bayes factor comparison was used to compare models and offered strong support for the variable rate models but could not distinguish between the two variable rate models (Supplementary Information), indicating that they are probably from the same congruent set of models for the species tree[82]. In subsequent discussion we primarily refer to results from the variable speciation rate model (for justification see Supplementary Information), although both variable rate models estimate similar patterns of net diversification rates through time (Supplementary Information).

Lineage-specific diversification rate estimation was performed in BAMM[83] and RevBayes. For analyses in BAMM, the setBammPriors function from the R package BAMMtools[84] was used to define appropriate priors. Different sets of analyses were performed with the prior for the expected number of shifts set to either 10 or 100. These different prior settings had minimal effect on parameter estimates. Clade-specific sampling fractions were specified for each sampled family and a backbone sampling fraction of 1 was used. We therefore accounted for incomplete sampling within families alongside comprehensive sampling of the backbone of the tree. For analyses in RevBayes, the dnCDBDP function was used and the prior for the total number of rate shifts was set to either 10 or 100. Clade-specific sampling fractions cannot be specified with this function. Therefore, the sampling fraction was set to 1 meaning that estimates will become inaccurate toward the present because of unsampled within-family diversity.

## Simulations on gene tree conflict

Simulations were based on a multispecies coalescent process. Each species tree contained 100 tips and was simulated as a birth–death branching process with time-dependent rates of speciation and extinction. In experiment 1, the extinction rate was always 0. The speciation rate was 0.75 species Myr$^{-1}$ at times over 6 Ma, between 6 and 2 Ma the speciation rate was 0.075 species Myr$^{-1}$ and less than 2 Ma the speciation rate was 0.75 species Myr$^{-1}$. In experiment 2, the net diversification rates were the same as in experiment 1; however, in this case changes to the extinction rate led to the net diversification rate shifts. Therefore, for all time intervals, the speciation rate was 0.75 species Myr$^{-1}$. At times

over 6 Ma the extinction rate was 0 species $Myr^{-1}$, between 6 and 2 Ma the extinction rate was 0.675 species $Myr^{-1}$ and at times less than 2 Ma the extinction rate was 0 species $Myr^{-1}$.

Species trees with extinct lineages have extra complexities: first, changes in the extinction rate have a less direct impact on the duration of extant lineages in the species tree compared to changes in the speciation rate (Supplementary Information); and second, the effect of extinction is reduced at times close to the present. This causes shorter branches in the species tree, leading to the so-called 'pull of the present'. We therefore performed a further analysis that was similar to experiment 2 but with no decrease in the extinction rate at the present. This offered insight into the effect of the 'pull of the present' on inferences of gene tree conflict and diversification rates and the relationship between these variables and the timing of rate shifts.

One-hundred gene trees were simulated along the branches of the birth–death branching processes according to a multispecies coalescent process. For most experiments, the effective population size was 5,000. In one further experiment, which was otherwise the same as experiment 1, the effective population size was 50,000. For each simulated dataset, the degree to which the simulated gene trees exhibited conflicting topologies with the species tree was plotted through time (Supplementary Information). This enabled characterization of the relationship between gene tree conflict caused by incomplete lineage sorting and shifts in speciation and extinction rates in the species tree.

More methods, results and discussion are available (Supplementary Information; Supplementary Figs. 13–24 and Supplementary Table 8).

### Inclusion and ethics statement

The research described here results from a highly inclusive, large-scale, international collaboration, which has actively encouraged the participation of many individuals from around the world. The authorship comprises many nationalities and is representative in terms of gender, career stage and career path. A total of 163 herbaria from 48 countries provided samples and/or house herbarium vouchers related to samples used in the study (see Acknowledgements). These samples originated from many countries, including Indigenous lands. We recognize the complex histories underlying all natural history collections and the global challenge we face in acknowledging them. We gave priority to recently collected samples and, as a result, most (85%) date from the postcolonial era (estimated here as 1970 onward). To share the benefits of our research, all data generated through this collaboration have been made publicly available before the submission of this work in several data releases, starting in 2019 (see Data availability).

### Reporting summary

Further information on research design is available in the Nature Portfolio Reporting Summary linked to this article.

## Data availability

All raw DNA sequence data generated for this study are deposited in the European Nucleotide Archive under the following bioprojects PRJNA478314, PRJEB35285, PRJEB49212 and PRJNA678873. All analysed data and metadata are available in Zenodo at https://doi.org/10.5281/zenodo.10778206 (ref. 55). The resulting trees and metadata are also available in GBIF (https://doi.org/10.15468/4njn8b) and Open Tree of Life (https://tree.opentreeoflife.org/curator/study/view/ot_2304). The names used in this work match the World Checklist of Vascular Plants (https://doi.org/10.34885/jdh2-dr22).

## Code availability

The code used and developed to perform analyses is available in GitHub at https://github.com/RBGKew/AngiospermPhylogenomics and Zenodo at https://doi.org/10.5281/zenodo.10778206 (ref. 55).

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

**Acknowledgements** The PAFTOL project was funded by grants from the Calleva Foundation to the Royal Botanic Gardens, Kew. Data were also contributed by the Genomics for Australian Plants Framework Initiative consortium funded by Bioplatforms Australia (enabled by the National Collaborative Research Infrastructure Strategy) and partner organizations. The work was further supported by research grants from VILLUM FONDEN (grant no. 00025354) and the Aarhus University Research Foundation (grant no. AUFF-E-2017-7-19) to W.L.E. and from grant nos NSF DBI 1930030 and DEB 1917146 to S.A.S. Computational resources and technical support were provided by the Research/Scientific Computing teams at The James Hutton Institute and the National Institute of Agricultural Botany (NIAB) through the 'UK's Crop Diversity Bioinformatics HPC' (BBSRC grant no. BB/S019669/1). The following provided technical assistance to the project at various stages: O. Berry, N. Black, M. Corcoran, S. Dequiret, I. Fairlie, L. Frankel, T. Freeth, A. Gilbert, B. Lepschi, D. Lewis, L. May, A. McArdle, E. O'Loughlin, S. Phillips, T. Sarkinen, L. Simmons, N. Walsh and M.-H. Weech. We thank all institutions who made their biological collections available and the many botanists and co-workers in the field who have collected, identified and curated the specimens used in this project. Specifically, we thank the following herbaria and their staff for providing samples for genomic analysis and/or for housing voucher specimens associated with analysed samples: A, ABH, AD, ALTB, APSC, B, BA, BC, BCN, BCRU, BG, BH, BHCB, BISH, BJFC, BKF, BM, BNRH, BOL, BONN, BR, BRI, BRIT, BRLU, BRUN, C, CAN, CANB, CAS, CBG, CNS, COL, CONC, CORD, CS, CTES, CUVC, DNA, E, EA, F, FI, FLAS, FMB, FTG, G, GB, GC, GENT, GH, GOET, GUAY, GZU, HAW, HEID, HITBC, HNG, HO, HPUJ, HRCB, HTW, HUA, HUAL, HUAZ, HUB, HUEFS, HUFU, IBSC, IBUG, ICN, IEB, INB, INPA, JBB, JBL, JRAU, K, KAS, KLU, KRB, KUN, L, LE, LISC, LP, LPB, LYJB, M, MA, MAU, MBA, MBML, MEDEL, MEL, MELU, MHA, MICH, MIN, MJG, MO, MSUN, MT, MY, N, NBG, NCU, NCY, NE, NH, NHM, NHMR, NMNL, NOU, NSW, NU, NY, OS, OSBU, P, PERTH, PG, PH, PRE, PTBG, QBG, QCA, QRS, RB, REU, S, SALA, SAR, SEV, SGO, SI, SING, SP, SPF, SPFR, SUVA, TCD,

TEX, TNS, TUH, TUM, U, UAPC, UB, UBT, UDW, UEC, UPCB, UPR, UPS, UPTC, US, USM, W, WAG, WS, WTU, YA and ZSS; acronyms follow Index Herbariorum (https://sweetgum.nybg.org/science/ih/). We also thank the Millennium Seed Bank Partnership for supporting access to samples. We acknowledge all national, state and regional authorities who authorized and facilitated the sourcing of these specimens. See also extended acknowledgements in the Supplementary Information.

**Author contributions** A.R.Z., T.C., A.A., S. Bellot, D.M.C., O.M.G., P.J.K., I.J.L., H. Sauquet, S.A.S., W.L.E., F. Forest and W.J.B were involved in conceptualization of this work. A.R.Z., T.C., A.A., S. Bellot, D.M.C., O.M.G., H. Sauquet, S.A.S., W.L.E., F. Forest and W.J.B. contributed to the methodology. A.R.Z., O.M., E.F., C. McGinnie, S.R.R., L. Simpson, J.J.C., R.S.C., S.D., L. Pokorny, G.M.A., K.G., K.P.H., A. Hoewener, A.-Q.H., E.M.J., I.A.B.S.K., I.L., D.A.L., E. J. Lírio, J.-X.L., P. Malakasi, N.A.S.P., T.S., J.V., G.K.A., R.L.A., M.S.A., M.A., M.J.A., J.A., W.A., J.B.B., C.D.B., H.F.B., M.D.B., R.L.B., R.J.B., M.J.B., E. Biffin, N.B., J.L.B., D.B., R.B., A.M.C.B., P. C. Boyce, G.L.C.B., M. Briggs, L. Broadhurst, G.K.B., J.J.B., A.B., S. Buerki, E. Burns, M. Byrne, S. Cable, A.C., M. W. Callmander, Á.C., D.J.C., W.M.C.-M., M.M.C., A.J.A.C., A.C.M., M. W. Chase, L.W.C., M.C., S. Chen, M.J.M.C., P.-A.C., M.A.C., S.C.C., J.G.C., X.C., T.L.P.C., I.D.C., L.C., I.D., G.D., N.M.J.D., A.P.D., K.-J.D., S.R.D., M.F.D., M.R.D., S.L.E., U.E., R.H.J.E., M. Escudero, M. Estrella, F. Fabriani, M.F.F., P.L.F., S.Z.F., R.M.F., S.F., L.F., T.F., M.G.-C., E.M.G., D.A.G., A. Giaretta, M.G., L.J.G., C.C.G., D.J.G., S.W.G., A. Grall, L.G., B.F.G., D.G.G., J.H., T. Haevermans, A. Haigh, J.C.H., T. Hall, M.J.H., S.A.H., O.H., T.R.H., G.D.H., H.C.F.H., C.J.J., S.A.J., R.W.J., G.K., I.M.K., K.K., E.A.K., G.J.K., B.K., B.B.K., R.R.K., S.K., M.A.K., J.H.L.-M., F.L., C.J.L., É.L.-B., G.P.L., D.-Z.L., L.L., S.L.-S., T.L., D.L., M. Lu, P.L.-I., J.L., E. J. Lucas, M. Luján, M. Lum, T.D.M., C. Magdalena, V.F.M., L.E.M., S.J.M., K. McColl, A.J.M., A.E.M., T.G.B.M., H.M., R.I.M., V.S.F.T.M., F.A.M., J.D.M., A.K.M., M.J.M., T.L.M., K. Mummenhoff, J.M., P. Muriel, D.J.M., K.N., L.N., F.J.N., R. Nyffeler, A.O., E.M.O., L. Palazzesi, A.L.P., S.K.P., J.P., D.S.P., O.A.P.-E., C. Persson, M.P., Y.P., J.R.P., G.M.P., R.F.P., G.T.P., C. Puglisi, M.Q., R.K.R., P.E.J.R., M. Renner, E.H.R., M. Rodda, Z.S.R., S.R., R.R., M.F.S., H. Schaefer, R. J. Schley, A.S.-L., A.S., I.S., K.A.S., M.P.S., A.O.S., A.R.G.S., M.S., E.C.S., J.F.S., N.S., D.E.S., P.S.S., R. J. Soreng, C.A.S., J.R.S., P.F.S., S.C.K.S., L. Struwe, J.M.T., I.R.H.T., A.H.T., I.T., A.T.-B., F.U., T.M.A.U., J.C.V., G.A.V., H.P.V., M.S.V., J.M.V., N.W., M.W., C.A.D.W., A.J.W., J.J.W., L.T.W., T.C.W., S.Y.W., L.A.W., R.W., S.W., M.X., Y.Y., Y.-X.Z., M.-Y.Z., S.Z., F.O.Z., S. Bellot, D.M.C., O.M.G., H. Sauquet, W.L.E., F. Forest and W.J.B. provided resources. A.R.Z., T.C., O.M., P. C. Bailey, K.L., G.E.B., N.E., E.F., B.G.-P., C. McGinnie, S.R.R., L. Simpson, L. Botigué, J.J.C., R.S.C., S.D., M.G.J., J.T.K., L. Pokorny, N.J.W., G.M.A., L.D., K.G., K.P.H., A. Hoewener, A.-Q.H., E.M.J., I.A.B.S.K., I.L., D.A.L., E. J. Lírio, J.-X.L., P. Malakasi, N.A.S.P., T.S., J.V. and S. Bellot carried out the investigations. A.R.Z., T.C., O.M., G.E.B., N.E., E.F., B.G.-P., C. McGinnie, R. Negrão, S.R.R., L. Simpson, E.T.R., V.M.A.B., K.P.H., J.V., T.R.A. and H. Sauquet were responsible for data curation. A.R.Z., T.C., P. C. Bailey and K.L. conducted the formal analysis. A.R.Z., T.C., P. C. Bailey, K.L., M.G.J. and J.T.K. developed the software. A.R.Z., T.C. and R. Negrão prepared the visualizations. A.R.Z., T.C., W.L.E., F. Forest and W.J.B. wrote the original manuscript with support from A.A., S. Bellot, D.M.C., O.M.G., H. Sauquet and S.A.S. K.L., E.F., J.J.C., J.T.K., L. Pokorny, N.J.W., A.-Q.H., E.M.J., I.L., J.V., M.S.A., J.B.B., M.D.B., R.L.B., A.M.C.B., L. Broadhurst, A.B., D.J.C., M.M.C., M. W. Chase, L.W.C., M.J.M.C., P.-A.C., T.L.P.C., U.E., R.H.J.E., M. Estrella, M.F.F., P.L.F., M.G., L.J.G., S.W.G., J.H., T. Haevermans, J.C.H., O.H., T.R.H., K.K., E.A.K., M.A.K., F.L., C.J.L., D.-Z.L., S.L.-S., T.D.M., V.S.F.T.M., F.A.M., A.K.M., M.J.M., D.J.M., F.J.N., L. Palazzesi, J.P., D.S.P., O.A.P.-E., Y.P., G.M.P., R.K.R., R.R., H. Schaefer, A.S.-L., M.P.S., A.R.G.S., N.S., D.E.S., P.S.S., R. J. Soreng, P.F.S., S.C.K.S., A.H.T., T.M.A.U., J.M.V., J.J.W., T.C.W., Y.Y., S.Z. and I.J.L. reviewed the final manuscript. S.A.S., W.L.E., F. Forest and W.J.B. undertook supervision. P.J.K., I.J.L., F. Forest and W.J.B. acquired funding. V.M.A.B., P.J.K., I.J.L., F. Forest and W.J.B. were responsible for project administration.

**Competing interests** The authors declare no competing interests.

**Additional information**
**Correspondence and requests for materials** should be addressed to William J. Baker.

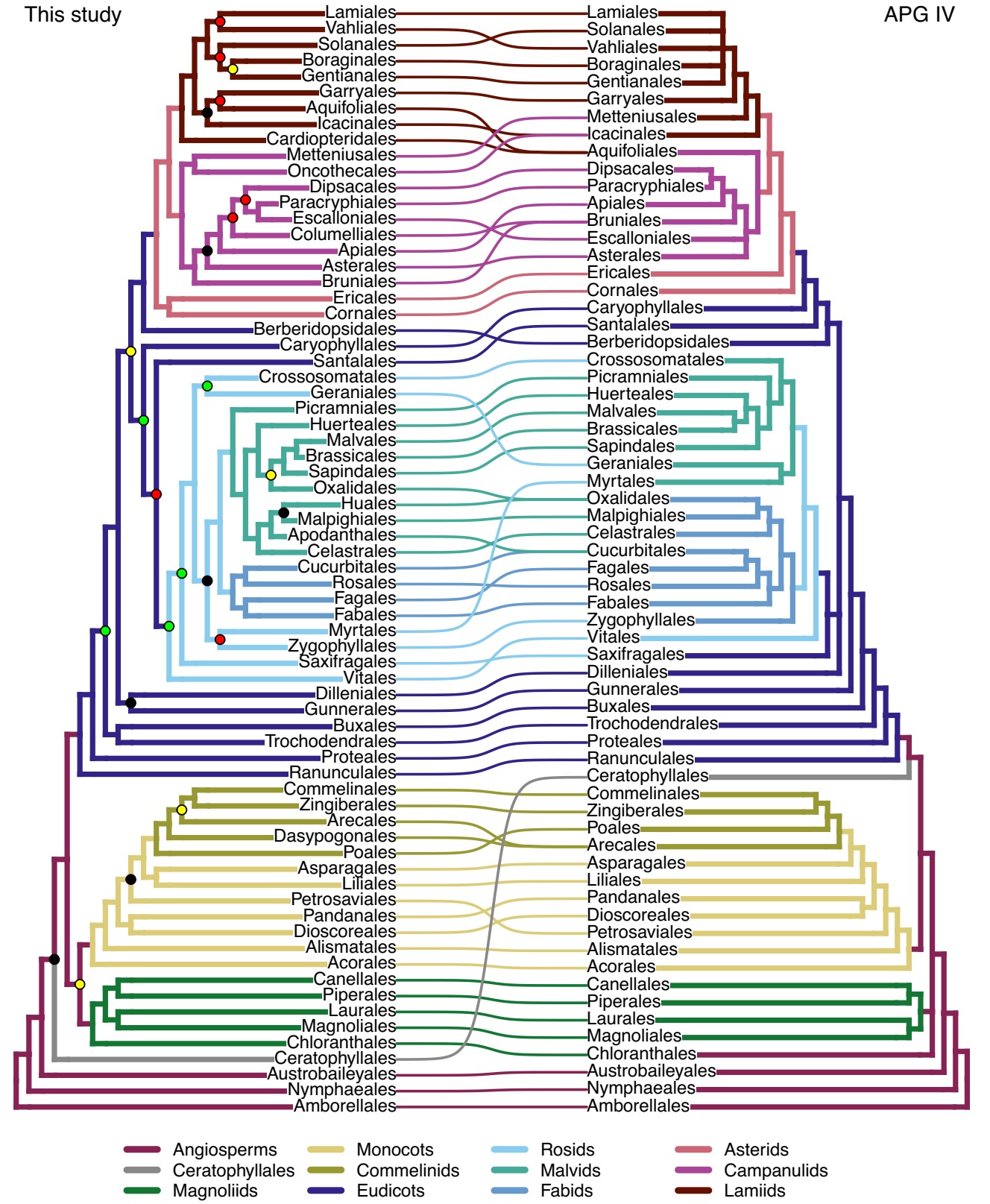

**Extended Data Fig. 1 | Tanglegram at ordinal level between this work (left) and the APG IV schematic tree (right).** Branches colours represent the clades according to the composition proposed in each work. Posterior probability is presented only for nodes without maximum support. Coloured circles in the left tree represent the posterior probability of each node as: maximum (absent), between 1 and 0.95 (green), between 0.95 and 0.75 (yellow), between 0.75 and 0.5 (red), below 0.5 (black).

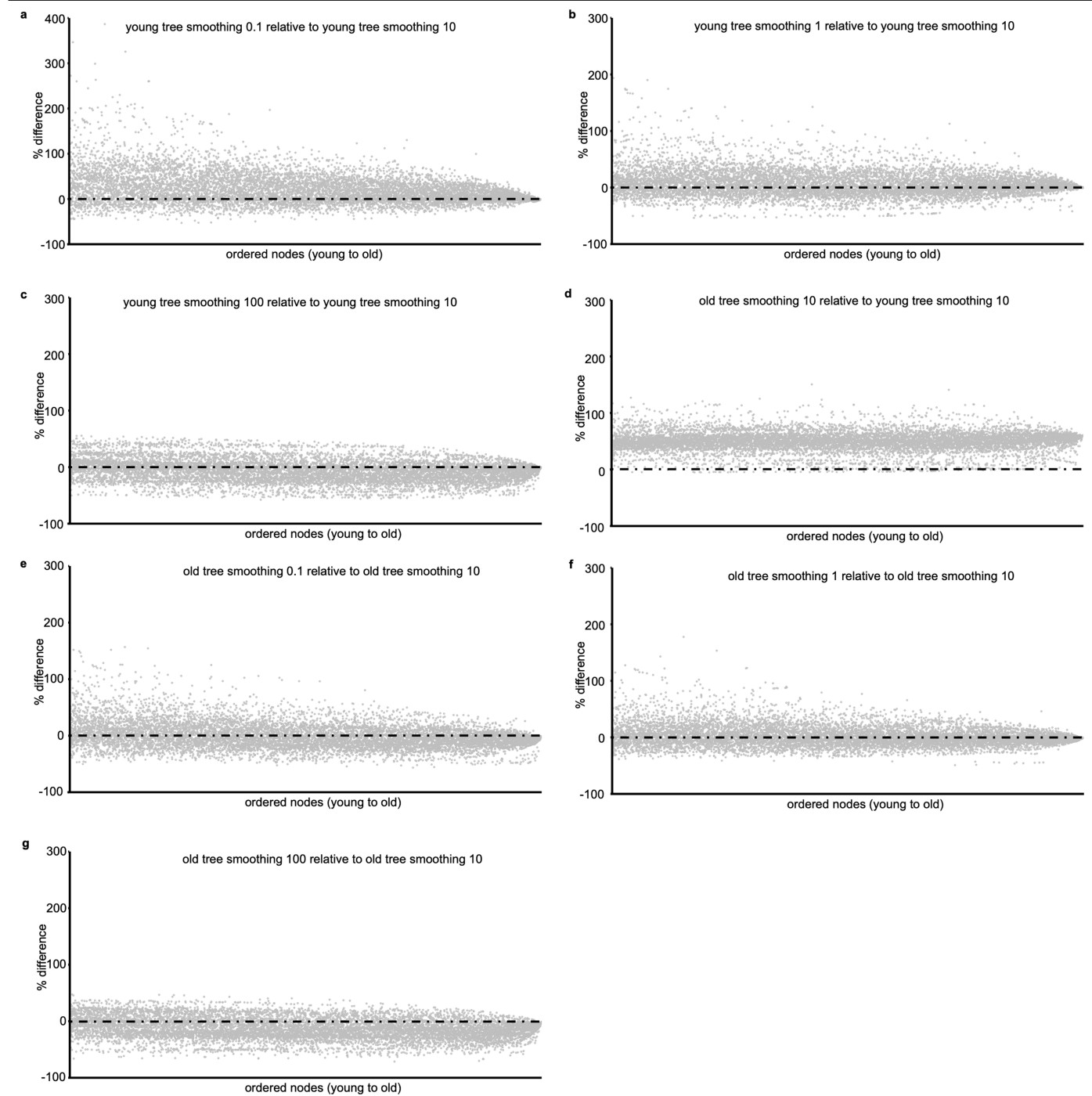

**Extended Data Fig. 2 | Comparison of node age estimates in the eight time-calibrated phylogenetic trees.** Each point represents a node and corresponds to the percentage difference in age estimates for that node between the two trees that are compared in each plot.

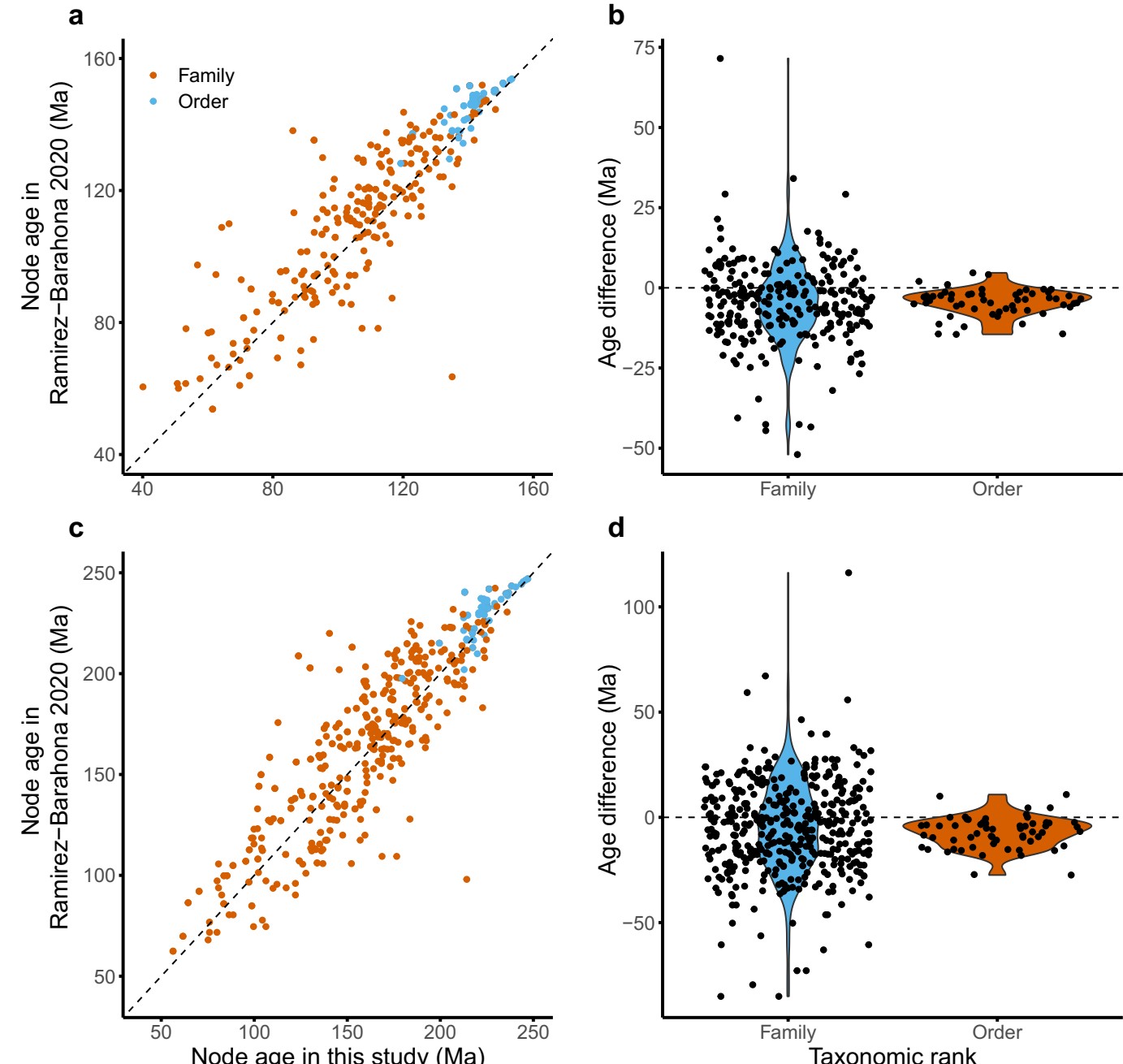

**Extended Data Fig. 3 | Comparison of stem ages of families and orders inferred in this study and Ramírez-Barahona et al.[5]. a** and **b**, Stem age comparison between our young tree (maximum constraint at the root node of 154 Ma) and the dataset CC_complete of Ramírez-Barahona et al.[5]. **a**, Ages in each study, coloured according to taxonomic rank and **b**, Age differences, calculated as age in this study minus age in Ramírez-Barahona et al.[5] **c** and **d**, Stem ages comparison between our old tree (maximum constraint at the root node of 247 Ma) and the dataset UC_complete from Ramírez-Barahona et al.[5] **c**, Ages in each study, coloured according to taxonomic rank and **d**, Age differences, calculated as age in this study minus age in Ramírez-Barahona et al.[5].

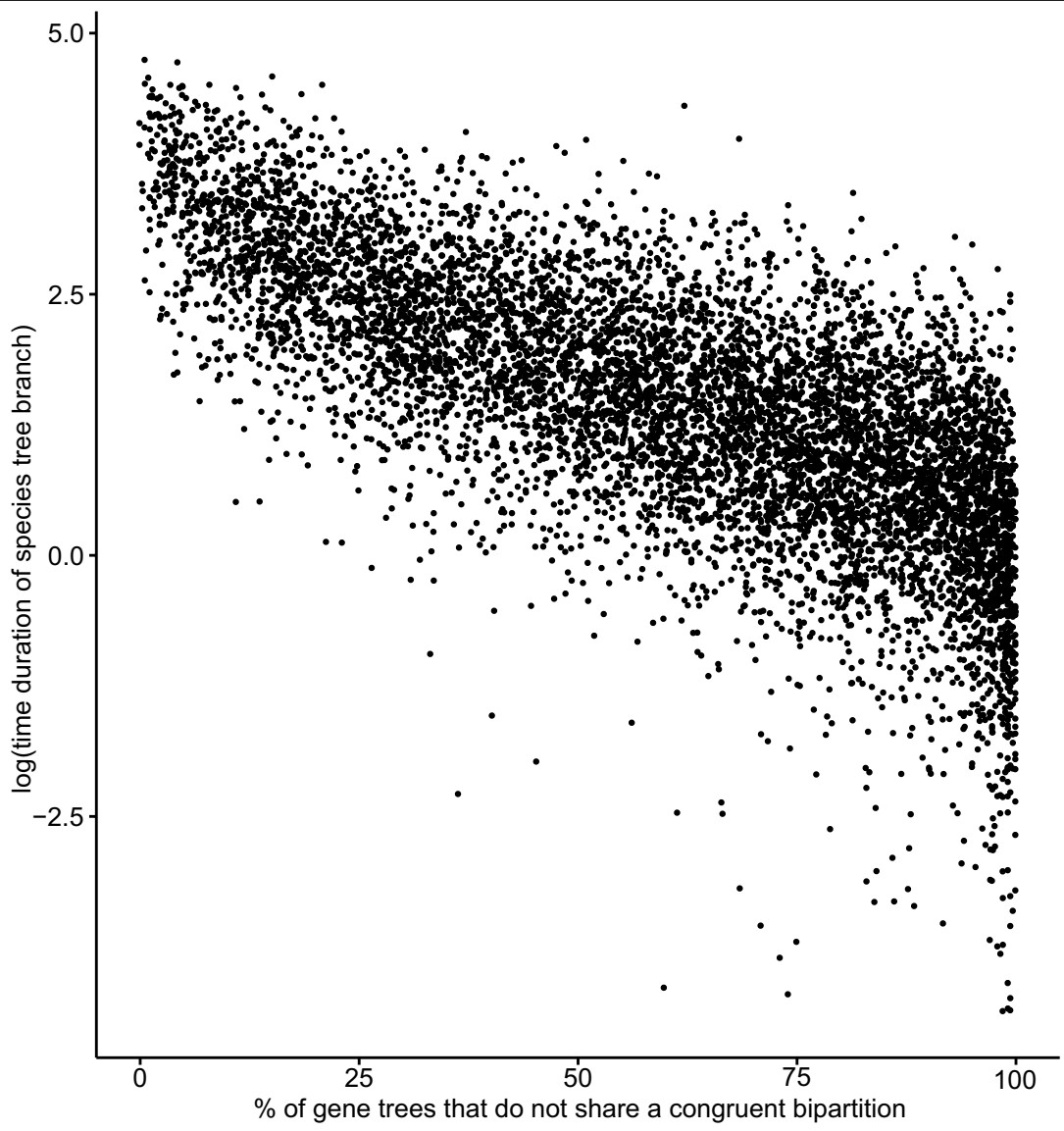

**Extended Data Fig. 4 | Correlation between branch time duration and percentage of gene trees that do not share a congruent bipartition for the branch.** The results are based on the young tree (maximum constraint at the root node of 154 Ma). For each branch in the young tree, the percentage of gene trees that do not share a congruent bipartition with the species tree branch is plotted against the logarithm of the time duration for the branch.

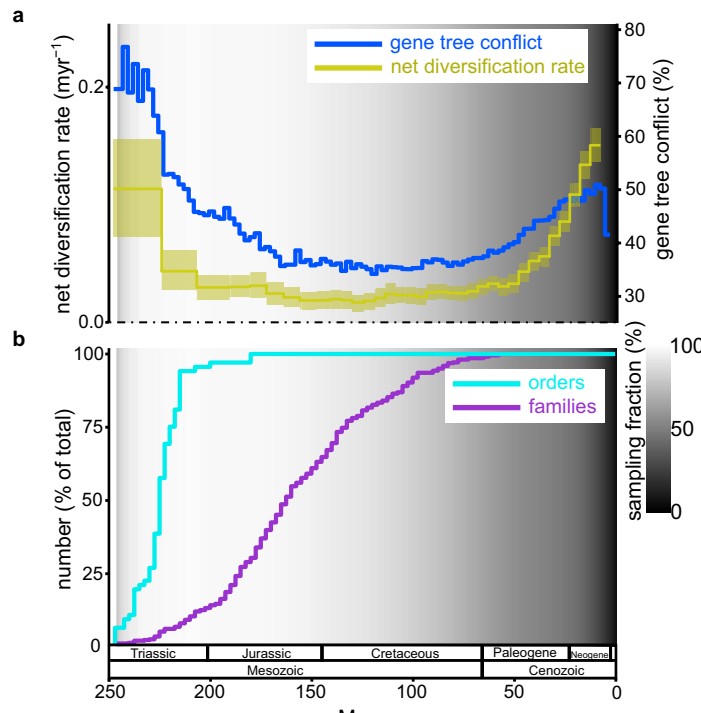

**Extended Data Fig. 5 | Angiosperm-wide diversification and gene tree conflict through time.** This is equivalent to Fig. 3 but for the old tree (maximum constraint at the root node of 247 Ma). **a**, Estimated net diversification rate through time (yellow, left y-axis) and the level of gene tree conflict through time (blue, right y-axis). Net diversification rates are estimated with a model that enables speciation rates to vary between time intervals; the line is the posterior mean and the yellow shaded area is the 95% highest posterior density. Gene tree conflict is calculated from the percentage of gene trees that do not share a congruent bipartition with each species tree branch, with the plotted value being the mean across all species tree branches that cross each 2.5 Myr time slice. **b**, Cumulative percentage of extant orders and families that have originated through time. In both a and b, the background grey-scale gradient is the estimated percentage of extant lineages represented in the species tree through time ("sampling fraction").

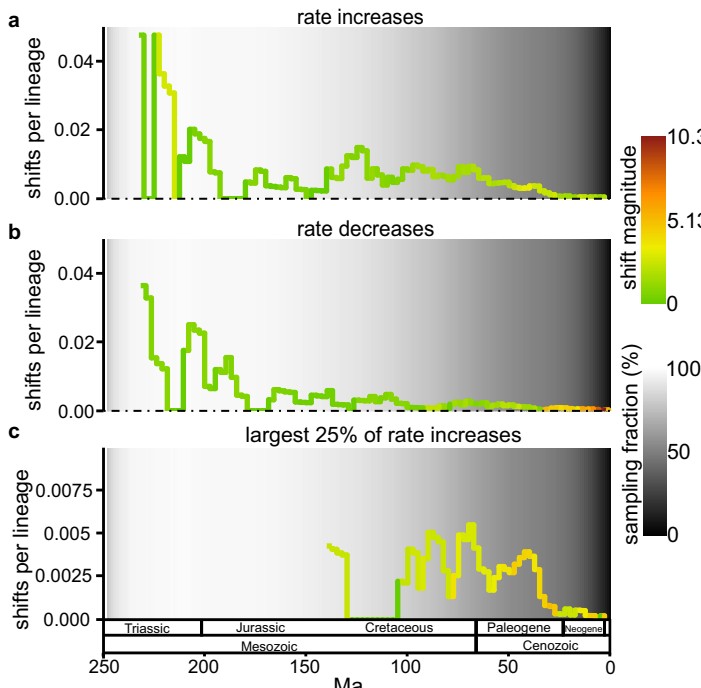

**Extended Data Fig. 6 | Summary of lineage-specific diversification rate shifts estimated by BAMM.** This is equivalent to Fig. 4, but for the old tree (maximum constraint at the root node of 247 Ma). **a**, Diversification rate increases per lineage through time. The colour corresponds to the average magnitude of the rate increases during the time period. **b**, Equivalent to a, but for rate decreases. **c**, Equivalent to a, but focusing on the largest 25% of diversification rate increases. In **a**, **b** and **c**, the number of shifts is extracted from the maximum a posteriori shift configuration, the prior for the number of shifts is set to 10 and the background grey-scale gradient is the estimated percentage of extant lineages represented in the species tree through time ("sampling fraction").

|---|---|

# Reporting Summary

## Statistics

For all statistical analyses, confirm that the following items are present in the figure legend, table legend, main text, or Methods section.

| n/a | Confirmed | |
|---|---|---|
| ☒ | ☐ | The exact sample size (*n*) for each experimental group/condition, given as a discrete number and unit of measurement |
| ☒ | ☐ | A statement on whether measurements were taken from distinct samples or whether the same sample was measured repeatedly |
| ☐ | ☒ | The statistical test(s) used AND whether they are one- or two-sided<br>*Only common tests should be described solely by name; describe more complex techniques in the Methods section.* |
| ☒ | ☐ | A description of all covariates tested |
| ☒ | ☐ | A description of any assumptions or corrections, such as tests of normality and adjustment for multiple comparisons |
| ☐ | ☒ | A full description of the statistical parameters including central tendency (e.g. means) or other basic estimates (e.g. regression coefficient) AND variation (e.g. standard deviation) or associated estimates of uncertainty (e.g. confidence intervals) |
| ☐ | ☒ | For null hypothesis testing, the test statistic (e.g. *F*, *t*, *r*) with confidence intervals, effect sizes, degrees of freedom and *P* value noted<br>*Give P values as exact values whenever suitable.* |
| ☐ | ☒ | For Bayesian analysis, information on the choice of priors and Markov chain Monte Carlo settings |
| ☒ | ☐ | For hierarchical and complex designs, identification of the appropriate level for tests and full reporting of outcomes |
| ☒ | ☐ | Estimates of effect sizes (e.g. Cohen's *d*, Pearson's *r*), indicating how they were calculated |

*Our web collection on statistics for biologists contains articles on many of the points above.*

## Software and code

Policy information about availability of computer code

| Data collection | No software was used. |
|---|---|
| Data analysis | Raw reads were trimmed using trimmomatic 0.39. Data recovery was performed in HybPiper 1.3.1, which includes blast 2.5.0, exonerate 2.4.0, parallel 20200922 and SPADES 3.13.0. Phylogenetic inference was performed using ASTRAL-III 5.7.3, ASTRAL-MP 5.15.5, fasttree 2.1.10, IQ-Tree 2.2.0-beta, MAFFT v7.480, newick_utils 1.6, phyutility 2.7.1, SortaDate commit 15bbbc4, treePL 1.0, Treeshrink 1.3.9. The summary of the alignments was produced with AMAS 1.0. The diversification analyses were performed with BAMM 2.5.0 and RevBayes 1.2.1.<br><br>The complete code used and developed to perform analyses is available in Zenodo (https://doi.org/10.5281/zenodo.10778206). |

For manuscripts utilizing custom algorithms or software that are central to the research but not yet described in published literature, software must be made available to editors and reviewers. We strongly encourage code deposition in a community repository (e.g. GitHub). See the Nature Portfolio guidelines for submitting code & software for further information.

# Data

Policy information about availability of data

All manuscripts must include a data availability statement. This statement should provide the following information, where applicable:
- Accession codes, unique identifiers, or web links for publicly available datasets
- A description of any restrictions on data availability
- For clinical datasets or third party data, please ensure that the statement adheres to our policy

All raw DNA sequence data generated for this study are deposited in the European Nucleotide Archive under the following bioprojects PRJNA478314, PRJEB35285, PRJEB49212 and PRJNA678873. All analysed data and metadata are available in Zenodo (https://doi.org/10.5281/zenodo.10778206). The resulting trees and metadata are also available in GBIF (https://doi.org/10.15468/4njn8b) and Open Tree of Life (https://tree.opentreeoflife.org/curator/study/view/ot_2304). The names used in this work match the World Checklist of Vascular Plants (https://doi.org/10.34885/jdh2-dr22).

# Research involving human participants, their data, or biological material

Policy information about studies with human participants or human data. See also policy information about sex, gender (identity/presentation), and sexual orientation and race, ethnicity and racism.

| | |
|---|---|
| Reporting on sex and gender | Not applicable |
| Reporting on race, ethnicity, or other socially relevant groupings | Not applicable |
| Population characteristics | Not applicable |
| Recruitment | Not applicable |
| Ethics oversight | Not applicable |

Note that full information on the approval of the study protocol must also be provided in the manuscript.

# Field-specific reporting

Please select the one below that is the best fit for your research. If you are not sure, read the appropriate sections before making your selection.

☐ Life sciences   ☐ Behavioural & social sciences   ☒ Ecological, evolutionary & environmental sciences

For a reference copy of the document with all sections, see nature.com/documents/nr-reporting-summary-flat.pdf

# Ecological, evolutionary & environmental sciences study design

All studies must disclose on these points even when the disclosure is negative.

| | |
|---|---|
| Study description | We sampled nearly 8,000 genera and 353 genes to infer a phylogenetic tree for angiosperms, dated and calibrated it with a dataset of 200 fossils, and used this evolutionary time frame to study the diversification of the group. |
| Research sample | We aimed to produce a dataset with at least one species per genus for >50% of the ca. 13,600 currently angiosperms genera. For genera with multiple species available, we retained up to three species, selecting primarily by phylogenetic representation followed by amount of data (number of genes and total length of recovery). Twelve species of gymnosperms were used as outgroups, one from each family. The final dataset contains 9,506 species, 7,923 genera, 416 families and 64 orders of angiosperms. It comprises 7,561 samples with data produced by this project and 1,963 samples mined from public repositories. |
| Sampling strategy | We sourced samples from herbarium, living, DNA bank and and tissue bank collections. |
| Data collection | We produced target sequence capture data using the Angiosperms353 probe kit. Total DNA was extracted using CTAB protocol and quantified by fluorometry. Average fragment size was assessed with 1% agarose gel. DNA extracts were diluted to 4 ng/ul. For extracts with high molecular weight, total DNA was fragmented using Covaris M220 Focused ultrasonicator. Genomic DNA libraries were prepared using NEBNext Ultra II DNA Library Prep Kit, following manufacturer's protocol at half volume, and with dual indexing. Libraries were normalised to 10 nM and pooled in equimolar amounts according to the average fragment size and taxonomic groups. Pools included, on average, 20-24 libraries. The pools were hybridised with Angiosperms353 probe kit. Hybridised pools were normalised to 7nM and combined for sequencing. Sequencing was carried out in Illumina platforms MiSeq and HiSeq. |
| Timing and spatial scale | Data production started in May 2016 and ended in December 2021. |
| Data exclusions | We excluded samples that failed both phylogenetic and barcode validation, as described in Baker (2022). Syst. Biol. 71: 301-319. We |

| Data exclusions | also excluded samples if more than three accessions were available for the same genus. |
| Reproducibility | All raw data and intermediate files are provided, making every step reproducible. All accessions used are listed in the Supplementary Table 1. |
| Randomization | Not applicable - randomization was not required. |
| Blinding | Not applicable - this work does not involve trials or controlled experiments. |

Did the study involve field work? ☐ Yes ☒ No

# Reporting for specific materials, systems and methods

We require information from authors about some types of materials, experimental systems and methods used in many studies. Here, indicate whether each material, system or method listed is relevant to your study. If you are not sure if a list item applies to your research, read the appropriate section before selecting a response.

## Materials & experimental systems

| n/a | Involved in the study |
|---|---|
| ☒ | ☐ Antibodies |
| ☒ | ☐ Eukaryotic cell lines |
| ☒ | ☐ Palaeontology and archaeology |
| ☒ | ☐ Animals and other organisms |
| ☒ | ☐ Clinical data |
| ☒ | ☐ Dual use research of concern |
| ☐ | ☒ Plants |

## Methods

| n/a | Involved in the study |
|---|---|
| ☒ | ☐ ChIP-seq |
| ☒ | ☐ Flow cytometry |
| ☒ | ☐ MRI-based neuroimaging |

