## [Peer Review File · Nature]

Manuscript Title: Phylogenomics and the rise of the angiosperms

Reviewer Comments & Author Rebuttals

Reviewer Reports on the Initial Version:

Referee expertise:

Referee #1: plant evolution

Referee #2: plant evolutionary genomics

Referee #3: angiosperm evolution

Referee #4: plant phylogenomics

Referees' comments:

Referee #1 (Remarks to the Author):

I really applaud this multi-authored and highly impressive collaboration. It is inspiring to see such a willingness to combine data and expertise in pursuit of ambitious evolutionary goals. The results are impressive and there is more than enough that is original here to justify publication. The application of target sequence capture to herbarium material is a significant innovation and has very substantially increased the extent of sampling in this hugely diverse clade (8,000 genera almost 60% of all living genera is amazing!). The paper is clearly appropriate for - and worthy of - publication in Nature.

I make a few comments below for the authors to consider in making modest revisions to the manuscript that might help to make it even more useful, and perhaps more balanced. I think some small changes just a sentence or two here and there would make this paper more helpful in setting out an expansive agenda for future research.

In general, and I am displaying my own, bias here, a little more acknowledgment of the value of paleobotanical data in addressing the whole question of angiosperm diversification would be valuable. It would be easy enough to cite some of the more comprehensive reviews as well as those that deal with the key original data. For example, I don't think it is so helpful on line 444 to simply cite Buggs and Darwin for "the sudden emergence of diverse angiosperm fossils during the Early Cretaceous." Neither is concerned with 20th century - let alone, 21st century - paleobotanical discoveries.

On line 253 it is emphasized that "limited and biased sampling of both taxa and genomes undermines confidence in the tree and its implications." The paper is thus set up as a response to this problem—by including 8,000 genes—about 60% of all angiosperm genera—and a set of 353 nuclear genes. This is very impressive. Importantly, the paper also asserts that it is a "critical test of earlier results." I agree with all of this – from my perspective the test really involves two sets of new perspectives/discoveries. One set concerns phylogenetic relationships – the other set concerns inferences on timing. I deal with these two aspects separately.

Relationships

I think it is not trivial that this paper "broadly supports" previous ideas of relationships. It is very encouraging that we are perhaps beginning to approximate some level of "evolutionary truth." Also

important is that the paper reveals novel relationships “and highlights some that remain intractable despite a vast increase in data”. Of the 64 orders recovered by previous analyses, 58 are also supported by this dataset. This is also the case for 406 of the 416 families recovered in previous work. However, there are some interesting novelties, especially among asterids, including the non-monophyly of Asteraceae. Also, interesting is that the position of Saxifragales, and Vitales with respect to other rosids is reversed. And from my paleobotanical perspective this is interesting since we have numerous reliable records of Saxifragales in the Cretaceous whereas the record of Vitales, based on their very distinctive seeds, are so far restricted to the Cenozoic. These phylogenetic adjustments are very valuable contribution of this paper.

Timing

The tree is scaled with 200 fossils based on a modified AngioCal dataset which is set out in the Supplementary Material. A small issue is in that data set I see only one data point of 127 million years or older, yet I see six dots - stated to represent fossil calibration points - in the figure legend that are in the Jurassic. I think this needs adjusting or explaining. Also, as a general point, I think it is important to indicate on figures with geological ages at least the Early versus Late Cretaceous boundary. The Cretaceous is such a long (and critical) interval that just portraying it as a single unit makes it difficult to understand these figures. Where the size of the figure permits, a more detailed breakdown into Cretaceous stages would be worthwhile.

I understand why the authors worked with an “old tree” at a “young tree,” although it would not add more than a few words to the paper to say that the “old tree” is dramatically a variance with the paleobotanical record. Similarly, the predictions from the “old tree” on the antiquity of eudicots, and some of its major groups, are especially hard to reconcile with a paleobotanical data. Very relevant to this point, and an observation frequently made in the paleobotanical literature, is that the sequence of appearance of angiosperm fossils in the early Cretaceous is entirely consistent with the phylogenetic pattern recognized by this and previous studies. What we have when angiosperms first appear is Nymphaeales, Austrobaileyales, Chloranthales and also Magnoliales. In the very earliest record, there is no evidence of eudicots, and when triaperturate pollen does appear, it is rare and very low in diversity. There is absolutely no indication of derived eudicots. If there were diverse eudicots in the Jurassic, why do they wait so long to appear in the fossil record and why do they not come in a much more haphazard way rather than in an orderly way that is consistent with your and other phylogenetic inferences?

You say “More controversially, the old tree places the early burst in the Triassic (Supplementary Fig. 9), highlighting that the scarcity of early angiosperm fossils for constraining the timeline of angiosperm diversification remains a major obstacle, regardless of the availability of genetic data.” Actually, I think it’s quite remarkable how much has been learned about the early angiosperm record from the Early Cretaceous – and as things stand right now, I think the record is already telling us quite a lot, but it points to the Early Cretaceous - not towards the Triassic.

Sampling

I took a careful look at the extant taxon sampling, and I think the authors have done a really terrific job in representing angiosperm diversity - even in relatively low diversity groups. For example in Chloranthaceae all four extant genera are represented. They have made an outstanding effort to represent generic diversity extremely well.

Nevertheless, I do want to make one final point about sampling, which I believe is important. It is true, as you say, in the abstract, “that limited and biased sampling of both taxa and genomes undermines our confidence in the tree and its implications.” Your remarkable paper takes us forward very significantly by the concerted effort to sample well and more could be done at the species level. But I doubt that it will change the picture in the earliest phases of angiosperm evolutionary history. The point I would make, without wishing to undermine in anyway the

importance of what has been accomplished, is that it is also important to step back and think outside the framework of extant taxa.

The sample of living taxa available to us - even if we were to sample every species - is itself limited, and no doubt biased in ways we do not fully understand, in the context of all of the angiosperm species that have ever lived over the more than 100 million year history of angiosperm evolution. As a famous paleontologist, one said, most species that have ever existed are extinct. The massive span of angiosperm evolutionary time has surely been marked by massive amounts of extinction, as well as spectacular diversification. This should be acknowledged, even if only very briefly. I recognize that in a sense this is a philosophical question, because it will be impossible to sample that diversity using molecular techniques. However, that diversity did exist and therefore it is not really helpful (or justifiable) to ignore it. It also begs the question of where we go from here.

As you say "Our sampling across angiosperms ensures that deeper branching events leading to extant lineages are comprehensively represented. However, our dated trees are sparsely sampled at the species-level, meaning that branching events are incompletely represented towards the present, limiting diversification inferences in that time-window." I would argue that with respect to deeper branching events you have come up against a similar problem - not so different from the one you are concerned about with species-level sampling and its implications in the Cenozoic - but with the big difference that the sampling cannot be extended in any meaningful way because the relevant taxa are extinct. In this area, frustrating as it may be, extant sampling has reached its limits. Of course, more species could be added, but you have captured most of the diversity already by your focus at the generic level.

In this context it is also interesting that high levels of conflict are "associated with an early burst of diversification". It is possible, that "The poor support for relationships among magnoliids, monocots, eudicots and Ceratophyllales might be explained by ancient hybridisation events, such as that recently proposed for the origin of the monocots." Not that you can do anything about it, but might part of the problem be poor sampling? I agree that "These examples highlight the importance of areas of poor resolution as waymarkers to biological events meriting further study." I believe that the most productive approach will be to pursue even more sophisticated molecular analyses, but in combination with a clear-eyed view of the crucial Early Cretaceous record. Some of the future research should seek to flesh out the missing diversity, at least in morphological terms, so that we can see what we are missing.

I offer these comments for the authors to consider. I do not regard any of them as compromising the importance of this paper, but I do think a few additional references and a few additional caveats could make this paper of even broader interest as we think about how we will tackle the question of angiosperm diversification in the future. In my view it has to be a twin pronged attack involving even more creativity in analysis of dramatically burgeoning amounts of molecular data, alongside new paleobotanical discoveries and research of the highest caliber.

Referee #2 (Remarks to the Author):

The authors are presenting a particularly extensive analysis of angiosperms' evolution based on the capture and sequencing of a set of 353 nuclear gene markers. This study includes sequences for new species, performing large phylogenetic reconstructions, and estimating time divergences. They show that the use of nuclear markers is largely recapitulating phylogenies based on plastid genes, with some marked differences. The divide-and-conquer approach the authors elicited for phylogenetics, incrementing the complexity of the reconstructions while minimizing the noise and computational burden, seems sound and robust in regards with the volume of data. The obtained tree for 8,000 genera is the largest available using nuclear markers. The resource is of high

interest, but the evolutionary insights appear rather general.

They notably used a backbone tree to guide subsequent analyses, but it is, as far as we can tell despite the automatic renaming of each file from the submission platform, not provided. While we appreciate that the list of species is included in sup table 1, considering the importance of this tree for the entire analysis, its inclusion as a supplementary material seems critical.

From our understanding, in order to build such backbone tree, the authors performed gene trees, which they then combined into a supertree using ASTRAL. How does this compare to a supergene approach, combining the global gene alignments instead of their trees?

The authors described several methodological decisions that should be justified:

The method section mentions that the trees were based on the GTR+G model. What is the rationale for this choice? If a test was performed to check this model's fitness, it should be mentioned.

Why 30% and 10% as the low support values for ultrafast bootstrap estimations and SH test respectively, below which branches are collapsed? These values seem extremely low, considering how these supports are usually interpreted (such as supported at $\geq 95\%$ for ultrafast bootstrap). What is the proportion of collapsed branches with the values currently used?

Some of the branches in Fig 1 are barely visible. The authors should consider a different background color to distinguish them better.

From the methodological side, the authors provide a comparison with the insights from plastid data, but it may be interesting to benchmark the approach more generally and its advantages/drawbacks compared with the transcriptomics efforts, and the potential future availability of whole genomes for a large part of the angiosperm tree. And it may also be of interest to know what new knowledge arises with this largest set compared to their previous efforts comprising >3000 species (Syst Biol. 2022 Feb 10;71(2):301-319).

Referee #3 (Remarks to the Author):

In this manuscript, a large team of plant evolutionary biologists report on the analysis of an immense dataset - 253 nuclear genes across over 9k species. The paper is beautifully written, clearly laying out the major findings, and very well illustrated with informative figures. I should note that I am not a phylogeneticist by training, so there could be some finer analytical points that I'm missing, but I think this is a lovely paper and am struggling to find anything to recommend for changes. They do an excellent job of laying out the caveats and enumerating the conflicts with previous large scale plastid studies. I wish I had some constructive criticisms but I can't find any!

Referee #4 (Remarks to the Author):

The authors present their manuscript, "Phylogenomics and the rise of the angiosperms." This is an unprecedented exploration of angiosperm phylogenetic relationships using the most richly sampled dataset to date both in terms of taxon and character sampling. This is largely possible due to the application of the angiosperm-wide sequence capture capabilities of the A353 probe set and the sampling of herbarium specimens; such a massive taxon sample with freshly collected material would have been nearly impossible (especially for endangered or now extinct species). While the results will be of major interest to the plant systematics, evolution, and ecology communities, I feel that this will be of much broader interest to those studying animal, fungal, and microbial systems (across myriad disciplines) given the enormous impact flowering plants have had on the history of life on Earth over the last ~150 million years.

The main contributions are: 1) a nearly comprehensive analysis of patterns of diversification over time in a clade that has perhaps shaped the history of life of Earth more than any other; and 2) major advances in taxonomy, systematics, and understanding of flowering plant biodiversity, corroborating many previous studies but also identifying some disagreements and areas that remain recalcitrant. The divergence time and diversification analyses are well executed, given the

enormity of the data analyzed, and clearly discussed in both the main document and the supplementary materials. The potential limitation (despite the astounding taxon sampling) is a relative lack of representation at the species level but the authors explicitly acknowledge this. Someday we will reach this point, but this endeavor is certainly a major leap in the right direction. I particularly appreciated the investigation of the relationship between diversification rates and gene tree conflict (they are strongly positively correlated), which I think will have major implications for the whole of evolutionary biology. Coalescent theory predicts this relationship, but to my knowledge, this is the first mega-scale demonstration of this phenomenon.

I do have a few comments/concerns about how the phylogenetic analyses were conducted, given that the major findings and implications of the paper are predicated on these (detailed below).

L255: "...build the tree of life..." "...infer the phylogenetic relationships..." would be more appropriate.

L260: Elevated relative to what? It should be enough to just say "...characterised by both gene tree conflict and..."

L264: "...clarify long-disputed..." It clarifies many but not all relationships.

L276: Not to mention fungi and prokaryotes.

L284: I disagree with the argument that the plastid genome is somehow more "readily accessible" than any other (nuclear, mitochondrial) genome. Perhaps the authors refer to the high relative copy number of the plastomes vs. nuclear loci, and the relative ease with which "genome skimming" approaches can allow the assembly of complete plastid genomes. Or, perhaps, it is an issue of historical reliance on genes and regions of the plastid genome due to the ease with which PCR primers could be designed, given this genome's highly conserved structure across angiosperms.

L289: "...address [hot topics] in plant evolutionary biology..." One could interpret "hot topics" to be something trendy or novel, though I see what the authors are trying to say here. This is among the most fundamental questions in plant systematics, so it should be conveyed as such. I think it would be more engaging to say something along the lines of: We now have the tools and resources to address fundamental questions on the origins of angiosperms in unprecedented ways, especially given that we are losing biodiversity at an alarming rate.

L295: Suggested rewording "...effectively capturing evolutionary divergences..." "...providing the most high-resolution estimates of divergence times to date..."

L298-301: This reads as if we know, with absolute certainty, what happened over millions of years. This is not the case. It would be more appropriate to frame the findings based on evidence. e.g. "We find [evidence for] high levels of conflict associated with..."

L306-307: "...total [known] angiosperm species diversity..."

L309-312: The authors have done a nice job emphasizing the importance of natural history collections in phylogenomics; conducting the fieldwork to accomplish such a massive taxon sample would have been prohibitive. However, many of these samples were initially collected from former colonies or indigenous lands and may be interpreted by some to represent a form of biotic or even cultural exploitation. It would be good to see the authors address this, even if only briefly, in the main manuscript body.

L317-331: I have some questions and concerns about the way the main phylogenomic analyses were conducted. I completely understand that analyzing a dataset of this magnitude represents an

immense computational undertaking, but in the end, I think it is a bit concerning that such an approach was taken, as in my view it does not harness the full power of the data.

"First, we computed a backbone tree, with sampling limited to five, phylogenetically representative species per family, totalling [sic] 15% (1,336) of all samples." What does "phylogenetically representative" mean here, specifically? Upon what information is this based? My concern here is that such an approach may introduce bias into the analysis by inadequate taxon sampling based on a priori selected representatives. It would have been informative to test some alternative sub-sampling schemes to assess the robustness to such an approach, thereby demonstrating that the current approach is warranted and unbiased.

"We used the backbone tree to guide the subsetting of our data for the construction of order-level alignments for each gene, which were then merged into global alignments." Were these subalignments then realigned globally? I could see this as another source of potential bias that could reinforce the effects on a priori (taxon-based) data partitions.

"We then computed global gene trees in order to generate a global species tree within a multispecies coalescent framework. To achieve this, we used our backbone gene trees as constraints to inform the computation of global gene trees, thus reducing tree space while still letting gene trees differ from each other. The smaller number of samples in the backbone dataset permits a more thorough search of tree space, resulting in greater confidence at deeper nodes than could be achieved in an unconstrained global analysis. This approach allows a trade-off between comprehensive sampling and tree search robustness while accommodating putative discordance among gene trees."

I am not completely clear on how this was accomplished, and I found no additional information in the supplementary methods. Has such an approach been used elsewhere successfully, and if so, was it tested with simulations and/or empirical data? "...we used our backbone gene trees as constraints to inform the computation of global gene trees, thus reducing tree space while still letting gene trees differ from each other." Then it would appear that the species tree estimate is not truly "global" in the sense that a full exploration of tree space was not conducted. I think a supplementary figure or illustration on the exact workflow of the authors' approach here would go a long way, and may further provide guidance to other researchers investigating mega-diverse clades with genomic data. Maybe I am misinterpreting here, but it seems to me that if the authors spent so much time and so many resources collecting such a massive dataset, then it makes little sense to restrict the phylogenetic analysis in such a way (which forms the basis for all downstream inference) without robustly assessing whether their approach is warranted and unbiased.

L337: The authors of ASTRAL-Pro claim that inclusion of multi-copy paralogs (given a mapping file) actually improves phylogenetic inference over methods that do not account for paralogy.

Chao Zhang, Celine Scornavacca, Erin K Molloy, Siavash Mirarab, ASTRAL-Pro: Quartet-Based Species-Tree Inference despite Paralogy, *Molecular Biology and Evolution*, Volume 37, Issue 11, November 2020, Pages 3292–3307, <https://doi.org/10.1093/molbev/msaa139>

L362: It has become common parlance in phylogenetic studies to use terms like "the phylogeny," "our phylogeny," "phylogenies," etc., to refer to the results of phylogenetic analyses. These are not correct, nor are they reflective of biological reality. There is "phylogeny," which represents the relationships of all organisms, extant and extinct, and encapsulates all of the complexities and intricacies of speciation, extinction, introgression, horizontal gene transfer, etc. A phylogenetic tree is a representation of biological and historical reality, and represents an estimate, an inference, and a hypothesis. Throughout, please be mindful of the terminology used: e.g. "Given their placement in our phylogen[etic analysis] and unique morphologies..."

Fig. 1. While this is a beautiful figure, the dates and names of each period are very small and faint.

Perhaps consider increasing their size and using abbreviations (e.g. 'J' for Jurassic). It may also be helpful to point out key clades with major diversification rate increases directly in the figure (i.e. name the clades or lineages in red).

L514-532: The Synthesis section does a nice job of summarizing the salient findings of this important work. However, "Our dated, genomic tree" [L525] may give an unwanted impression. Consider "Our fossil-calibrated, phylogenomic hypothesis of angiosperm relationships..."

--Craig Barrett

Author Rebuttals to Initial Comments:

**Responses to reviewers' comments on Nature manuscript
2023-09-16621A**

**Zuntini, Carruthers et al. "Phylogenomics and the rise of the
angiosperms"**

Referee #1:

I really applaud this multi-authored and highly impressive collaboration. It is inspiring to see such a willingness to combine data and expertise in pursuit of ambitious evolutionary goals. The results are impressive and there is more than enough that is original here to justify publication. The application of target sequence capture to herbarium material is a significant innovation and has very substantially increased the extent of sampling in this hugely diverse clade (8,000 genera almost 60% of all living genera is amazing!). The paper is clearly appropriate for - and worthy of - publication in Nature.

I make a few comments below for the authors to consider in making modest revisions to the manuscript that might help to make it even more useful, and perhaps more balanced. I think some small changes just a sentence or two here and there would make this paper more helpful in setting out an expansive agenda for future research.

>>> We are grateful to the reviewer for this positive feedback. We are delighted to hear this appreciation of the scale and significance of the phylogenomic work we have undertaken.

In general, and I am displaying my own, bias here, a little more acknowledgment of the value of paleobotanical data in addressing the whole question of angiosperm diversification would be valuable. It would be easy enough to cite some of the more comprehensive reviews as well as those that deal with the key original data. For example, I don't think it is so helpful on line 444 to simply cite Buggs and Darwin for "the sudden emergence of diverse angiosperm fossils during the Early Cretaceous." Neither is concerned with 20th century - let alone, 21st century - paleobotanical discoveries.

>>> The reviewer makes a good point here and, as a result, we have inserted the following additional references at that point in the manuscript to acknowledge palaeobotanical evidence more thoroughly:

Coiro M, Doyle JA, Hilton J. 2019. How deep is the conflict between molecular and fossil evidence on the age of angiosperms? *New Phytologist* 223: 83–99.

Herendeen PS, Friis EM, Pedersen KR, Crane PR. 2017. Palaeobotanical redux: revisiting the age of the angiosperms. *Nature Plants* 3: 17015.

Friis EM, Crane PR, Pedersen KR. 2011. *Early flowers and angiosperm evolution*. Cambridge University Press.

On line 253 it is emphasized that "limited and biased sampling of both taxa and genomes undermines confidence in the tree and its implications." The paper is thus set up as a response to this problem—by including 8,000 genes—about 60% of all angiosperm genera—and a set of 353 nuclear genes. This is very impressive.

Importantly, the paper also asserts that it is a “critical test of earlier results.” I agree with all of this – from my perspective the test really involves two sets of new perspectives/discoveries. One set concerns phylogenetic relationships — the other set concerns inferences on timing. I deal with these two aspects separately.

Relationships

I think it is not trivial that this paper “broadly supports” previous ideas of relationships. It is very encouraging that we are perhaps beginning to approximate some level of “evolutionary truth.” Also important is that the paper reveals novel relationships “and highlights some that remain intractable despite a vast increase in data”. Of the 64 orders recovered by previous analyses, 58 are also supported by this dataset. This is also the case for 406 of the 416 families recovered in previous work. However, there are some interesting novelties, especially among asterids, including the non-monophyly of Asteraceae. Also, interesting is that the position of Saxifragales, and Vitales with respect to other rosids is reversed. And from my paleobotanical perspective this is interesting since we have numerous reliable records of Saxifragales in the Cretaceous whereas the record of Vitales, based on their very distinctive seeds, are so far restricted to the Cenozoic. These phylogenetic adjustments are very valuable contribution of this paper.

>>> Thank you for these interesting observations.

Timing

The tree is scaled with 200 fossils based on a modified AngioCal dataset which is set out in the Supplementary Material. A small issue is in that data set I see only one data point of 127 million years or older, yet I see six dots - stated to represent fossil calibration points - in the figure legend that are in the Jurassic. I think this needs adjusting or explaining. Also, as a general point, I think it is important to indicate on figures with geological ages at least the Early versus Late Cretaceous boundary. The Cretaceous is such a long (and critical) interval that just portraying it as a single unit makes it difficult to understand these figures. Where the size of the figure permits, a more detailed breakdown into Cretaceous stages would be worthwhile.

>>> We have checked the plotting of the calibration points and they are correctly shown. The dots on Fig 1 denote the *phylogenetic placement* of the node calibrations, but the dates reflect the *resulting node age estimates* from the divergence time analysis, not the age of the fossil itself. We have added a clarification in the figure legend “Black dots at nodes indicate the phylogenetic placement of fossil calibrations based on the updated AngioCal fossil calibration dataset. Note that calibrated nodes can be older than the age of the corresponding fossils due to the use of minimum age constraints.” We have also revisited the timescale on our tree figures and have now indicated the boundaries of Early and Late Cretaceous, as requested.

I understand why the authors worked with an “old tree” at a “young tree,” although it would not add more than a few words to the paper to say that the “old tree” is dramatically a variance with the paleobotanical record. Similarly, the predictions from the “old tree” on the antiquity of eudicots, and some of its major groups, are especially hard to reconcile with a paleobotanical data. Very relevant to this point, and an observation frequently made in the paleobotanical literature, is that the sequence of appearance of angiosperm fossils in the early Cretaceous is entirely consistent with the phylogenetic pattern recognized by this and previous studies.

What we have when angiosperms first appear is Nymphaeales, Austrobaileyales, Chloranthales and also Magnoliales. In the very earliest record, there is no evidence of eudicots, and when triaperturate pollen does appear, it is rare and very low in diversity. There is absolutely no indication of derived eudicots. If there were diverse eudicots in the Jurassic, why do they wait so long to appear in the fossil record and why do they not come in a much more haphazard way rather than in an orderly way that is consistent with your and other phylogenetic inferences?

You say “More controversially, the old tree places the early burst in the Triassic (Supplementary Fig. 9), highlighting that the scarcity of early angiosperm fossils for constraining the timeline of angiosperm diversification remains a major obstacle, regardless of the availability of genetic data.” Actually, I think it’s quite remarkable how much has been learned about the early angiosperm record from the Early Cretaceous – and as things stand right now, I think the record is already telling us quite a lot, but it points to the Early Cretaceous - not towards the Triassic.

>>> Again, we are grateful for these interesting observations on our findings. As requested, we have edited the text to highlight that the old tree is dramatically at variance with the fossil record, as follows: “More controversially, the old tree places the early burst in the Triassic (Supplementary Fig. 9), which is dramatically at variance with the palaeobotanical record, highlighting the fact that current molecular dating methods are unable to resolve the age of angiosperms.”

Sampling

I took a careful look at the extant taxon sampling, and I think the authors have done a really terrific job in representing angiosperm diversity - even in relatively low diversity groups. For example in Chloranthaceae all four extant genera are represented. They have made an outstanding effort to represent generic diversity extremely well. Nevertheless, I do want to make one final point about sampling, which I believe is important. It is true, as you say, in the abstract, “that limited and biased sampling of both taxa and genomes undermines our confidence in the tree and its implications.” Your remarkable paper takes us forward very significantly by the concerted effort to sample well and more could be done at the species level. But I doubt that it will change the picture in the earliest phases of angiosperm evolutionary history. The point I would make, without wishing to undermine in anyway the importance of what has been accomplished, is that it also important to be step back think outside the framework of extant taxa.

The sample of living taxa available to us - even if we were to sample every species - is itself limited, and no doubt biased in ways we do not fully understand, in the context of all of the angiosperm species that have ever lived over the more than 100 million year history of angiosperm evolution. As a famous paleontologist, one said, most species that have ever existed are extinct. The massive span of angiosperm evolutionary time has surely been marked by massive amounts of extinction, as well as spectacular diversification. This should be acknowledged, even if only very briefly. I recognize that in a sense this is a philosophical question, because it will be impossible to sample that diversity using molecular techniques. However, that diversity did exist and therefore it is not really helpful (or justifiable) to ignore it. It also begs the question of where we go from here.

As you say “Our sampling across angiosperms ensures that deeper branching events leading to extant lineages are comprehensively represented. However, our dated trees are sparsely sampled at the species-level, meaning that branching events are incompletely represented towards the present, limiting diversification

inferences in that time-window.” I would argue that with respect to deeper branching events you have come up against a similar problem - not so different from the one you are concerned about with species-level sampling and its implications in the Cenozoic - but with the big difference that the sampling cannot be extended in any meaningful way because the relevant taxa are extinct. In this area, frustrating as it may be, extant sampling has reached its limits. Of course, more species could be added, but you have captured most of the diversity already by your focus at the generic level.

>>> We agree with these sentiments and have added a brief statement regarding angiosperm extinction, as requested, in the following sentence: “With our sampling across angiosperms, we ensured that deeper branching events leading to extant lineages are comprehensively represented, while recognising that extinct lineages are inaccessible to genomic methods.”

In this context it is also interesting that high levels of conflict are “associated with an early burst of diversification”. It is possible, that “The poor support for relationships among magnoliids, monocots, eudicots and Ceratophyllales might be explained by ancient hybridisation events, such as that recently proposed for the origin of the monocots.” Not that you can do anything about it, but might part of the problem be poor sampling? I agree that “These examples highlight the importance of areas of poor resolution as waymarkers to biological events meriting further study.” I believe that the most productive approach will be to pursue even more sophisticated molecular analyses, but in combination with a clear-eyed view of the crucial Early Cretaceous record. Some of the future research should seek to flesh out the missing diversity, at least in morphological terms, so that we can see what we are missing.

>>> Thank you for these exciting ideas on future research. We hope that our research sparks new activity in these directions.

I offer these comments for the authors to consider. I do not regard any of them as compromising the importance of this paper, but I do think a few additional references and a few additional caveats could make this paper of even broader interest as we think about how we will tackle the question of angiosperm diversification in the future. In my view it has to be a twin pronged attack involving even more creativity in analysis of dramatically burgeoning amounts of molecular data, alongside new paleobotanical discoveries and research of the highest caliber.

>>> We are very grateful to reviewer 1 for their positive and insightful review, and their interesting reflections on our work.

Referee #2:

The authors are presenting a particularly extensive analysis of angiosperms' evolution based on the capture and sequencing of a set of 353 nuclear gene markers. This study includes sequences for new species, performing large phylogenetic reconstructions, and estimating time divergences. They show that the use of nuclear markers is largely recapitulating phylogenies based on plastid genes, with some marked differences. The divide-and-conquer approach the authors elicited for phylogenetics, incrementing the complexity of the reconstructions while minimizing the noise and computational burden, seems sound and robust in regards with the volume of data. The obtained tree for 8,000 genera is the largest available

using nuclear markers. The resource is of high interest, but the evolutionary insights appear rather general.

They notably used a backbone tree to guide subsequent analyses, but it is, as far as we can tell despite the automatic renaming of each file from the submission platform, not provided. While we appreciate that the list of species is included in sup table 1, considering the importance of this tree for the entire analysis, its inclusion as a supplementary material seems critical.

>>> We apologise for omitting the backbone tree - this was an oversight. The backbone tree is now provided in the Supplementary Material (Supplementary Fig. 2).

From our understanding, in order to build such backbone tree, the authors performed gene trees, which they then combined into a supertree using ASTRAL. How does this compare to a supergene approach, combining the global gene alignments instead of their trees?

>>> As coalescent methods and ASTRAL in particular are widely acknowledged as the “state-of-the-art” and more desirable than concatenation analyses, we have opted for this approach here (Davidson et al., 2015: <https://doi.org/10.1186/1471-2164-16-S10-S1>). Furthermore, concatenation analyses could not be meaningfully performed on our full dataset for computational reasons.

The authors described several methodological decisions that should be justified:

The method section mentions that the trees were based on the GTR+G model. What is the rationale for this choice? If a test was performed to check this model's fitness, it should be mentioned.

>>> The GTR+G is a very flexible model that allows simpler nested models to emerge if the data support them and therefore model tests were not performed. Simulation studies also question the value of model-testing if a more flexible model can be used (Abadi et al., 2019: - <https://www.nature.com/articles/s41467-019-08822-w>).

Why 30% and 10% as the low support values for ultrafast bootstrap estimations and SH test respectively, below which branches are collapsed? These values seem extremely low, considering how these supports are usually interpreted (such as supported at $\geq 95\%$ for ultrafast bootstrap). What is the proportion of collapsed branches with the values currently used?

>>> We are following recommendations by the developers of the methods that we use, which are substantiated by analysis of simulated data (Mirarab, 2019: <https://arxiv.org/pdf/1904.03826.pdf>) and are routinely applied in phylogenomic analysis (e.g., Zhang et al., 2017: https://link.springer.com/chapter/10.1007/978-3-319-67979-2_4).

Some of the branches in Fig 1 are barely visible. The authors should consider a different background color to distinguish them better.

>>> Depicting trees at this scale for publication figures is exceptionally challenging. The reviewer's perception of the figure may also have been compromised by downsampled graphics in the review pdf. We have tried a wide variety of different colour schemes and backgrounds - the selected scheme was, by far, the best and is

consistent with comparable large trees depicted elsewhere (e.g., Kawahara et al., 2023: <https://www.nature.com/articles/s41559-023-02041-9>).

From the methodological side, the authors provide a comparison with the insights from plastid data, but it may be interesting to benchmark the approach more generally and its advantages/drawbacks compared with the transcriptomics efforts, and the potential future availability of whole genomes for a large part of the angiosperm tree. And it may also be of interest to know what new knowledge arises with this largest set compared to their previous efforts comprising >3000 species (Syst Biol. 2022 Feb 10;71(2):301-319).

>>> It would be difficult (and probably not as informative as one would expect) to undertake these comparisons due to the significant incompleteness of relevant studies relative to ours. We are not in favour of drawing comparisons to the tree in our 2022 paper as this was, ostensibly, a protocol paper rather than a “proper” tree paper. While a comparison to our 2022 paper may be interesting to some, we feel it would be off-topic in this case and would require a substantial extension to this manuscript. Moreover, such a comparison would not only reflect an increase in data, it would also be impacted by the evolution of our methods, compromising any meaningful interpretation.

Referee #3:

In this manuscript, a large team of plant evolutionary biologists report on the analysis of an immense dataset - 253 nuclear genes across over 9k species. The paper is beautifully written, clearly laying out the major findings, and very well illustrated with informative figures. I should note that I am not a phylogeneticist by training, so there could be some finer analytical points that I'm missing, but I think this is a lovely paper and am struggling to find anything to recommend for changes. They do an excellent job of laying out the caveats and enumerating the conflicts with previous large scale plastid studies. I wish I had some constructive criticisms but I can't find any!

>>> We are very grateful for these positive reflections on our work.

Referee #4:

The authors present their manuscript, “Phylogenomics and the rise of the angiosperms.” This is an unprecedented exploration of angiosperm phylogenetic relationships using the most richly sampled dataset to date both in terms of taxon and character sampling. This is largely possible due to the application of the angiosperm-wide sequence capture capabilities of the A353 probe set and the sampling of herbarium specimens; such a massive taxon sample with freshly collected material would have been nearly impossible (especially for endangered or now extinct species). While the results will be of major interest to the plant systematics, evolution, and ecology communities, I feel that this will be of much broader interest to those studying animal, fungal, and microbial systems (across myriad disciplines) given the enormous impact flowering plants have had on the history of life on Earth over the last ~150 million years.

The main contributions are: 1) a nearly comprehensive analysis of patterns of diversification over time in a clade that has perhaps shaped the history of life of Earth more than any other; and 2) major advances in taxonomy, systematics, and understanding of flowering plant biodiversity, corroborating many previous studies but also identifying some disagreements and areas that remain recalcitrant. The

divergence time and diversification analyses are well executed, given the enormity of the data analyzed, and clearly discussed in both the main document and the supplementary materials. The potential limitation (despite the astounding taxon sampling) is a relative lack of representation at the species level but the authors explicitly acknowledge this. Someday we will reach this point, but this endeavor is certainly a major leap in the right direction. I particularly appreciated the investigation of the relationship between diversification rates and gene tree conflict (they are strongly positively correlated), which I think will have major implications for the whole of evolutionary biology. Coalescent theory predicts this relationship, but to my knowledge, this is the first mega-scale demonstration of this phenomenon.

>>> Thank you for this positive feedback on our work.

I do have a few comments/concerns about how the phylogenetic analyses were conducted, given that the major findings and implications of the paper are predicated on these (detailed below).

L255: "...build the tree of life..." "...infer the phylogenetic relationships..." would be more appropriate.

>>> We prefer to retain the original wording in the interests of accessibility to the broad readership of *Nature*.

L260: Elevated relative to what? It should be enough to just say "...characterised by both gene tree conflict and..."

>>> We have changed "elevated" to "high". The suggested alternative is not adequate because gene tree conflict is all pervasive.

L264: "...clarify long-disputed..." It clarifies many but not all relationships.

>>> "Many" added as requested.

L276: Not to mention fungi and prokaryotes.

>>> This is a fair point, but this is not an exhaustive list. We have added "fungi".

L284: I disagree with the argument that the plastid genome is somehow more "readily accessible" than any other (nuclear, mitochondrial) genome. Perhaps the authors refer to the high relative copy number of the plastomes vs. nuclear loci, and the relative ease with which "genome skimming" approaches can allow the assembly of complete plastid genomes. Or, perhaps, it is an issue of historical reliance on genes and regions of the plastid genome due to the ease with which PCR primers could be designed, given this genome's highly conserved structure across angiosperms.

>>> We have changed "readily accessible" to "widely sequenced".

L289: "...address [hot topics] in plant evolutionary biology..." One could interpret "hot topics" to be something trendy or novel, though I see what the authors are trying to say here. This is among the most fundamental questions in plant systematics, so it should be conveyed as such. I think it would be more engaging to say something along the lines of: We now have the tools and resources to address fundamental questions on the origins of angiosperms in unprecedented ways, especially given that we are losing biodiversity at an alarming rate.

>>> We have changed “hot topics” to “fundamental questions”. While we appreciate the sentiment behind “*especially given that we are losing biodiversity at an alarming rate*” we feel it is not strictly relevant here.

L295: Suggested rewording “...effectively capturing evolutionary divergences...”
“...providing the most high-resolution estimates of divergence times to date...”

>>> We prefer and have retained our original wording.

L298-301: This reads as if we know, with absolute certainty, what happened over millions of years. This is not the case. It would be more appropriate to frame the findings based on evidence. e.g. “We find [evidence for] high levels of conflict associated with...”

>>> We accepted and incorporated this suggestion.

L306-307: “...total [known] angiosperm species diversity.”

>>> We accepted and incorporated this suggestion.

L309-312: The authors have done a nice job emphasizing the importance of natural history collections in phylogenomics; conducting the fieldwork to accomplish such a massive taxon sample would have been prohibitive. However, many of these samples were initially collected from former colonies or indigenous lands and may be interpreted by some to represent a form of biotic or even cultural exploitation. It would be good to see the authors address this, even if only briefly, in the main manuscript body.

>>> We are sympathetic to the points raised and following the recommendation of the senior editor Michelle Trenkman, we have added the following Inclusion and ethics statement.

“The research described here results from a highly inclusive, large-scale, international collaboration that has actively encouraged the participation of numerous individuals from around the world. The authorship comprises many nationalities and is representative in terms of gender, career stage and career path. A total of 163 herbaria from 48 countries provided samples and/or house herbarium vouchers related to samples used in the study (see Acknowledgements). These samples originated from numerous countries, including indigenous lands. We recognise the complex histories underlying all natural history collections and the global challenge that we face in acknowledging them. We prioritised recently collected samples and, as a result, the vast majority (85%) date from the post-colonial era (estimated here as 1970 onward). To share the benefits of our research, all data generated through this collaboration have been made publicly available prior to the submission of this work in several data releases, commencing in 2019 (see Data Availability).”

L317-331: I have some questions and concerns about the way the main phylogenomic analyses were conducted. I completely understand that analyzing a dataset of this magnitude represents an immense computational undertaking, but in the end, I think it is a bit concerning that such an approach was taken, as in my view it does not harness the full power of the data.

>>> We welcome feedback on the methods used and have endeavoured to address the points raised by the reviewer below. He is correct that the computational undertaking is truly immense, demanding substantial computational resource and time - our methodological approach is framed in that context. We have reflected carefully on all of his comments and have conducted some additional exploratory analyses, which we hope the reviewer will find reassuring.

“First, we computed a backbone tree, with sampling limited to five, phylogenetically representative species per family, totalling [sic] 15% (1,336) of all samples.” What does “phylogenetically representative” mean here, specifically? Upon what information is this based? My concern here is that such an approach may introduce bias into the analysis by inadequate taxon sampling based on a priori selected representatives. It would have been informative to test some alternative sub-sampling schemes to assess the robustness to such an approach, thereby demonstrating that the current approach is warranted and unbiased.

>>> We appreciate that this could have been more clearly explained in the original manuscript. Here, “phylogenetically representative” means that samples were selected to represent the earliest divergences in each family based on published phylogenetic evidence and preliminary analyses. The rationale for this was to ensure that the crown node of each family was represented as effectively as possible and thus to increase confidence in the inferred relationships among orders and families. These deep divergences must be properly characterised in backbone gene trees given their role as constraints in the global analysis. We have rephrased the sentence quoted above as follows: “First, we computed a backbone species tree with sampling limited to five species per family (1,336 [15%] samples in total) and targeted to represent their deepest nodes.” We have also clarified the method for doing this in the Methods section.

Following the reviewer’s request, we explored the potential bias that might be introduced by the choice of species included in the backbone tree. We inferred 20 replicates of the backbone, randomly selecting up to five samples per family (among the 50% best samples in terms of gene number and gene length recovered). We then summarised the trees to family level and computed Robinson-Foulds distances between the trees. We found some variation 1) among the replicates (mean: 4.87%) and 2) between the replicates and our backbone tree (mean: 4.72%). However, the nodes that varied among these trees were largely poorly supported anyway. Moreover, the distances to the original backbone are not significantly different (p -value = 0.3397; T-test) to the distances among replicates. These additional findings increase our confidence that our approach was reasonable. We have provided a new supplementary figure (Supplementary Fig. 1) and updated the methods accordingly: “To evaluate the extent to which sample selection might affect the backbone tree topology, we inferred 20 backbone replicates, randomly selecting five samples (among the 50% best samples in terms of gene number and gene length recovered). We then summarised the trees to family level and computed Robinson-Foulds distances between the backbone and the 20 replicates (see Supplementary Fig. 14).”

“We used the backbone tree to guide the subsetting of our data for the construction of order-level alignments for each gene, which were then merged into global alignments.” Were these subalignments then realigned globally? I could see this as

another source of potential bias that could reinforce the effects on a priori (taxon-based) data partitions.

>>> The sub-alignments were realigned globally across sub-alignments but not within sub-alignments. We have added a short clarifying sentence (“This approach yields alignment across the order-level sub-alignments without disrupting the structure within the sub-alignments.”). A similar approach has been used to infer the fern Tree of Life (Nita et al., 2023: <https://doi.org/10.3389/fpls.2022.909768>). All alignment methods in any phylogenetic analysis have their limitations, especially when working with a wide taxonomic sampling and fragmentary data. However, we believe that our approach alleviates more biases than it creates by seeking a trade-off in the challenge of aligning across taxonomic levels (exacerbated by fragmentary data). Other than biases that might affect any phylogenetic study, we cannot identify a specific bias of special concern to our study that we could have addressed in a proportionate manner, other than perhaps the extremely unlikely situation where a genus is placed within the wrong ordinal alignment.

“We then computed global gene trees in order to generate a global species tree within a multispecies coalescent framework. To achieve this, we used our backbone gene trees as constraints to inform the computation of global gene trees, thus reducing tree space while still letting gene trees differ from each other. The smaller number of samples in the backbone dataset permits a more thorough search of tree space, resulting in greater confidence at deeper nodes than could be achieved in an unconstrained global analysis. This approach allows a trade-off between comprehensive sampling and tree search robustness while accommodating putative discordance among gene trees.” I am not completely clear on how this was accomplished, and I found no additional information in the supplementary methods. Has such an approach been used elsewhere successfully, and if so, was it tested with simulations and/or empirical data? “...we used our backbone gene trees as constraints to inform the computation of global gene trees, thus reducing tree space while still letting gene trees differ from each other.” Then it would appear that the species tree estimate is not truly “global” in the sense that a full exploration of tree space was not conducted. I think a supplementary figure or illustration on the exact workflow of the authors’ approach here would go a long way, and may further provide guidance to other researchers investigating mega-diverse clades with genomic data. Maybe I am misinterpreting here, but it seems to me that if the authors spent so much time and so many resources collecting such a massive dataset, then it makes little sense to restrict the phylogenetic analysis in such a way (which forms the basis for all downstream inference) without robustly assessing whether their approach is warranted and unbiased.

>>> We really like the suggestion of the workflow diagram - we have now prepared such a diagram and included it in the supplementary data (Supplementary Fig. 1). We hope that this helps readers to understand the work more clearly.

The reviewer also questions the “globalness” of our study given that we applied some constraints to the analysis. We would argue that the analysis has a greater chance of achieving effective global accuracy by circumventing implausible parts of the treespace, given the scale of the computational task at hand. The constraint trees that we applied result from the backbone analysis, in which it was feasible to explore treespace extensively across major lineages to establish the broad picture. We wanted to ensure that the insights from the backbone analysis could positively

influence the global analysis, in which treespace exploration was so much more challenging. It is important to stress though that our constraints were really rather loose. Firstly, all constraint trees were collapsed using a highly conservative threshold of 80% bootstrap (we accidentally omitted this detail in the original manuscript - now corrected). Secondly, the constraints were applied on a gene-tree-by-gene-tree basis, resulting in independence across potentially contrasting gene histories. We have revised the relevant methods section for clarity as follows: “We used the gene trees from the backbone analysis to constrain the topology of each respective global gene tree. To avoid propagating error from the backbone analysis to the global analysis, we removed potentially misleading signal from the backbone gene trees prior to applying them as constraints. First, branches with bootstrap values below 80% were collapsed to avoid enforcing poorly supported relationships. Second, tips placed far from the rest of their order were algorithmically removed (but retained in global gene alignments).”

L337: The authors of ASTRAL-Pro claim that inclusion of multi-copy paralogs (given a mapping file) actually improves phylogenetic inference over methods that do not account for paralogy. Chao Zhang, Celine Scornavacca, Erin K Molloy, Siavash Mirarab, ASTRAL-Pro: Quartet-Based Species-Tree Inference despite Paralogy, *Molecular Biology and Evolution*, Volume 37, Issue 11, November 2020, Pages 3292–3307, <https://doi.org/10.1093/molbev/msaa139>

>>> We agree that accounting for paralogs is a desirable future direction of research, but it is out of scope for this study, due to the immense scale of the additional computational load that it would entail. We recognise the value of digging deeper into the detail here, but are reassured that the approach we have taken is valid and credible, in light of recent papers (e.g.: Yan et al., 2022: <https://doi.org/10.1093/sysbio/syab056>; Zhang & Mirarab, 2022: <https://doi.org/10.1093/molbev/msac215>).

L362: It has become common parlance in phylogenetic studies to use terms like “the phylogeny,” “our phylogeny,” “phylogenies,” etc., to refer to the results of phylogenetic analyses. These are not correct, nor are they reflective of biological reality. There is “phylogeny,” which represents the relationships of all organisms, extant and extinct, and encapsulates all of the complexities and intricacies of speciation, extinction, introgression, horizontal gene transfer, etc. A phylogenetic tree is a representation of biological and historical reality, and represents an estimate, an inference, and a hypothesis. Throughout, please be mindful of the terminology used: e.g. “Given their placement in our phylogen[etic analysis] and unique morphologies...”

>>> We agree. We have reviewed the document and edited it accordingly.

Fig. 1. While this is a beautiful figure, the dates and names of each period are very small and faint. Perhaps consider increasing their size and using abbreviations (e.g. ‘J’ for Jurassic). It may also be helpful to point out key clades with major diversification rate increases directly in the figure (i.e. name the clades or lineages in red).

>>> We have revised the geological timescales on all tree figures to improve clarity. In the interests of readability we have not added details of the key clades with major diversification rate increases. These can be fully studied in the expanded version of this tree available in the supplementary material (Supplementary Fig. 23).

L514-532: The Synthesis section does a nice job of summarizing the salient findings of this important work. However, “Our dated, genomic tree” [L525] may give an unwanted impression. Consider “Our fossil-calibrated, phylogenomic hypothesis of angiosperm relationships...”

>>> We rephrased this as follows: “Our fossil-calibrated, phylogenomic tree ...”

--Craig Barrett

Reviewer Reports on the First Revision:

Referees' comments:

Referee #1 (Remarks to the Author):

The authors have responded adequately to the reviews provided and the paper is now suitable for publication.

In my view, this paper should be taken as the "last word" - for now - on the phylogenetic relationships of major groups of angiosperms. I doubt very much whether additional sampling of extant taxa at the species level is going to change these results substantially, although it may impact some details and yield some insights into specific aspects of the Cenozoic diversification.

Much more interesting would be new results that are significantly different in some way from the current paradigm of phylogenetic patterns - that has emerged - with relatively little change - over the past couple of decades.

Referee #2 (Remarks to the Author):

The authors address my concerns adequately.

Referee #4 (Remarks to the Author):

I have re-read the manuscript, and reviewed the authors' responses to all of the reviews. I appreciate both the level of rigor in the original submission and the attention to detail in their revisions and responses to the reviews. This is especially true of my comments on using a backbone-tree approach: the authors conducted a sensitivity analysis to test the method (I'm glad to hear that there was not much of an effect of taxon selection!). I think this adds an interesting and important aspect to the manuscript, demonstrating that the results are sound, given the propensity and/or need for heuristic shortcuts with such massive datasets. Congratulations on a job well done to this long list of authors. I'm sure this will have a major impact once published, and I look forward to seeing it published (and to the news coverage that will likely follow). -- Craig Barrett